# Integrated analysis of H2A.Z isoforms function reveals a complex interplay in gene regulation

Assala Lamaa[1], Jonathan Humbert[2], Marion Aguirrebengoa[3], Xue Cheng[2], Estelle Nicolas[1], Jacques Côté[2], Didier Trouche[1]*

[1]LBCMCP, Centre de Biologie Intégrative, Université de Toulouse, CNRS, UPS, Toulouse, France; [2]St-Patrick Research Group in Basic Oncology, Laval University Cancer Research Center and Oncology Division of CHU de Québec-Université Laval Research Center, Quebec, Canada; [3]BigA Core Facility, Centre de Biologie Intégrative, Université de Toulouse, CNRS, UPS, Toulouse, France

**Abstract** The H2A.Z histone variant plays major roles in the control of gene expression. In human, H2A.Z is encoded by two genes expressing two isoforms, H2A.Z.1 and H2A.Z.2 differing by three amino acids. Here, we undertook an integrated analysis of their functions in gene expression using endogenously-tagged proteins. RNA-Seq analysis in untransformed cells showed that they can regulate both distinct and overlapping sets of genes positively or negatively in a context-dependent manner. Furthermore, they have similar or antagonistic function depending on genes. H2A.Z.1 and H2A.Z.2 can replace each other at Transcription Start Sites, providing a molecular explanation for this interplay. Mass spectrometry analysis showed that H2A.Z.1 and H2A.Z.2 have specific interactors, which can mediate their functional antagonism. Our data indicate that the balance between H2A.Z.1 and H2A.Z.2 at promoters is critically important to regulate specific gene expression, providing an additional layer of complexity to the control of gene expression by histone variants.

**\*For correspondence:**
didier.trouche@univ-tlse3.fr

**Competing interests:** The authors declare that no competing interests exist.

## Introduction

The H2A.Z histone variant is one of the two histone variants conserved from yeast to human. It is enriched at the −1 and +1 nucleosomes surrounding the nucleosome-depleted region of active promoters (*Barski et al., 2007*). It can play positive or negative roles in specific gene expression (*Subramanian et al., 2015*). In addition to transcriptional control, H2A.Z is important for genetic stability and DNA damage repair (*Billon and Côté, 2012*), although its exact function is somewhat controversial in mammals (*Taty-Taty et al., 2014*; *Xu et al., 2012*). H2A.Z is incorporated in chromatin by specialized machinery relying on the ATPase SWR1 in yeast and its orthologs in other species (*Billon and Côté, 2012*; *Mizuguchi et al., 2004*). In mammals, both SWR1 orthologs, SRCAP and p400 belong to multimolecular complexes that have been shown to mediate incorporation of the H2A.Z histone variant (*Gévry et al., 2007*; *Ruhl et al., 2006*). H2A.Z removal can be mediated by the ANP32E chaperon protein (*Obri et al., 2014*).

In vertebrates, H2A.Z is encoded by two different genes, *H2AFZ* and *H2AFV*, leading to the production of three proteins, one produced from the *H2AFZ* gene, called H2A.Z.1, and two splicing variants produced from *H2AFV*, called H2A.Z.2 and H2A.Z.2.2 (*Bönisch et al., 2012*; *Matsuda et al., 2010*). Importantly, the *H2AFZ* gene is essential for mouse development (*Faast et al., 2001*), and conditional deletion of *H2AFZ* in the brain leads to neurogenesis defects (*Shen et al., 2018*), highlighting the importance of H2A.Z in mammals. Intriguingly, H2A.Z.1 and H2A.Z.2 proteins are highly similar, differing by only three amino acids (*Dryhurst et al., 2009*; *Eirín-López et al., 2009*).

They are expressed at different levels as shown by the analysis of their expression in human embryonic and adult tissues (*Dryhurst et al., 2009*). It was recently found that they have overlapping function or can compensate each other since depleting both isoforms in the mouse intestine leads to severe homeostasis defects whereas individual mutants have no phenotype (*Zhao et al., 2019*). However, it was shown by ChIP-Seq analysis of exogenously expressed GFP-tagged isoforms that their genomic localisation is similar but not identical (*Pünzeler et al., 2017*). This could indicate that H2A.Z.1 and H2A.Z.2 functions could be divergent. Indeed, specific functions and/or expression of these isoforms in human cancer cells have been described. For example, despite the fact that the mRNA for both isoforms are overexpressed in human melanoma, only depletion of H2A.Z.2 was found to favour melanoma cell proliferation (*Vardabasso et al., 2015*). In contrast, H2A.Z.1 is specifically overexpressed in hepatocellular carcinoma (HCC) (*Yang et al., 2016*). Differences in the physiological roles of H2A.Z.1 and H2A.Z.2 have also been documented. In highly recombinogenic DT40 chicken cells, *Nishibuchi et al. (2014)* found that H2A.Z.2, but not H2A.Z.1, is recruited to and functionally important for the repair of double strand breaks *Dunn et al., 2017*). However, no study has extensively investigated the independent and interdependent roles of both isoforms in their physiological context in specific gene expression in non-transformed cells. Furthermore, the lack of specific antibodies, preventing the investigation of endogenous proteins, is a strong limitation to understand the molecular bases of these differences.

Here, we performed an integrated study of H2A.Z.1- and H2A.Z.2-dependent gene expression in untransformed cells by combining RNA-seq, RT-qPCR and ChIP assays. This unveiled the parallel and antagonistic functions of H2A.Z.1 and H2A.Z.2 in gene regulation. Furthermore, by tagging endogenous H2A.Z.1 and H2A.Z.2 isoforms, we were able to identify proteins interacting specifically with one isoform or the other. Our study thus reveals major antagonism between H2A.Z.1 and H2A.Z.2 regarding control of gene expression, mediated by specific interactors.

## Results

### H2A.Z.1 and H2A.Z.2 are major regulators of gene expression

In an effort to identify genes regulated by H2A.Z isoforms H2A.Z.1 and H2A.Z.2 in non-transformed cells, we transfected Telomerase-immortalized WI38 human primary fibroblasts with specific siRNAs silencing either H2A.Z.1, H2A.Z.2, or both at the same time. Analysis of each H2A.Z isoform mRNA expression by RT-qPCR indicated that both siRNAs are efficient and specific (*Figure 1A*), although transfection of the H2A.Z.2 siRNA slightly affected the expression of H2A.Z.1 mRNA. Using U2OS cells in which the two alleles coding for H2A.Z.1 or H2A.Z.2 were tagged with a 3xFlag-2xStrep tag by genome editing (see *Figure 1—figure supplement 1A* for the characterization of the cell lines) (*Dalvai et al., 2015*), we found that siRNA-mediated depletion efficiently decreased the expression of one isoform without affecting the expression of the other (see *Figure 1B* for a H2A.Z western blot and *Figure 1—figure supplement 1B* for a Flag western blot). Note that in these western blots, we observed a band migrating above H2A.Z and decreasing upon siRNA depletion. This band probably corresponds to a post-translational modification of H2A.Z, most likely its ubiquitination (see below). No obvious difference could be observed between H2A.Z.1 and H2A.Z.2 with respect to this band (*Figure 1—figure supplement 1B*). By performing a western blot analysis using an antibody recognising total H2A.Z, we found that the strong depletion of total H2A.Z in WI38 cells required transfection of both siRNAs together, as we observed only a moderate or weak effect upon H2A.Z.1 or H2A.Z.2 depletion, respectively (*Figure 1C*). Quantification of this experiment is consistent with the interpretation that total H2A.Z is composed of about 2/3rd of H2A.Z.1 and 1/3rd of H2A.Z.2 in WI38 cells (*Figure 1C*).

We next performed RNA-Seq experiments to identify the genes regulated by both isoforms in an unbiased fashion. Two entirely independent experiments were performed and differential analysis showed that the expression levels of 3573 mRNAs were significantly affected upon H2A.Z.1 depletion and 1500 upon H2A.Z.2 depletion (see *Supplementary files 1–4* for the list of deregulated genes). 41.5% and 50.4% of regulated genes were activated upon H2A.Z.1 and H2A.Z.2 depletion, respectively. In addition, 691 mRNAs were significantly affected only upon depletion of the two isoforms together (371 activated and 320 repressed) (*Supplementary file 5*), suggesting that at these promoters H2A.Z isoforms can compensate each other. These results indicate that H2A.Z.1 and

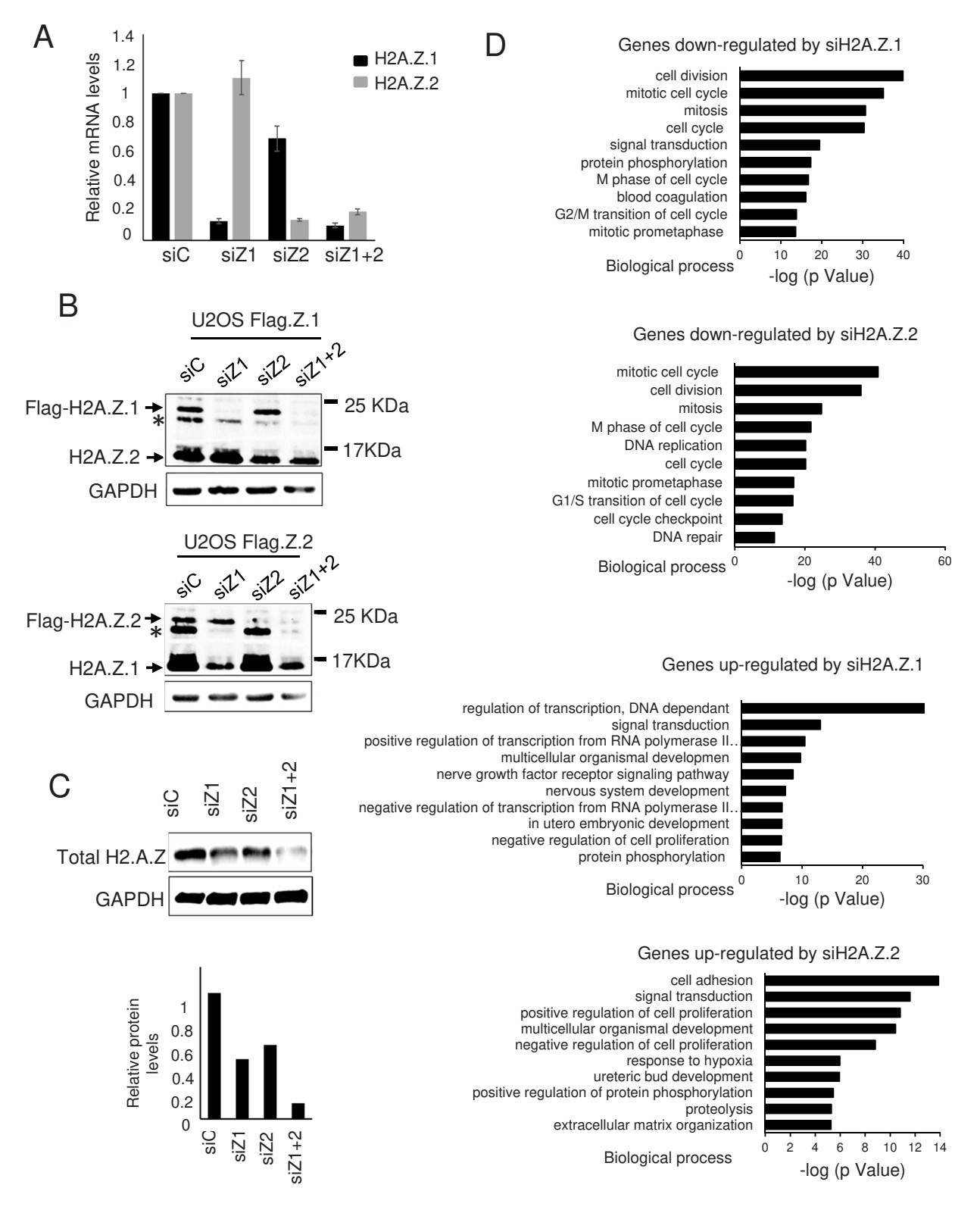

**Figure 1.** Effect of H2A.Z.1 and H2A.Z.2 depletion on gene expression. (**A**) WI38 cells were transfected with the indicated siRNAs. 72 hr later, total RNA was prepared. The amount of H2A.Z.1 and H2A.Z.2 mRNA was quantified by RT-qPCR, standardised using GAPDH mRNA levels and calculated relative to one for cells transfected with the control siRNA. The mean and SDOM from five independent experiments are shown. (**B**) Genome edited U2OS cells expressing either tagged H2A.Z.1 (top) or tagged H2A.Z.2 (bottom) were transfected with the indicated siRNAs. 72 hr later, total cell extracts were

*Figure 1 continued on next page*

*Figure 1 continued*

prepared and subjected to western blot analysis using an anti H2A.Z antibody. The star * indicates a band probably corresponding to a post-translationally modified untagged H2A.Z isoform. (C) Same as in A, except that total cell extracts were prepared and subjected to western blot analysis using the indicated antibody, then protein signals were standardised using GAPDH protein levels and calculated relative to1 for cells transfected with the control siRNA. A representative experiment out of two is shown. (D) Gene ontology analyses (Genecodis) of genes downregulated upon H2A.Z.1or H2A.Z.2 depletion or upregulated upon H2A.Z.1 or H2A.Z.2 depletion (from top to bottom). The top 10 most significant enrichments are shown.

The online version of this article includes the following source data and figure supplement(s) for figure 1:

**Source data 1.** Source data of the histogramme representing the depletion of H2A.

**Figure supplement 1.** Characterisation of U2OS cells genome-edited to express 3xFlag-2xStrep H2A.Z.1 or H2A.Z.2.

H2A.Z.2 are major regulators of gene expression in non-transformed cells, both acting as gene activators or repressors. Note that the expression of H2A.Z isoforms is strongly reduced but not abolished upon transfection of the siRNAs. We thus may have missed some H2A.Z-regulated genes in this analysis, when a residual amount of H2A.Z is sufficient to bring about the correct regulation.

Gene ontology analyses indicate that genes activated upon H2A.Z1 knock-down are enriched in negative regulators of cell proliferation whereas genes repressed are mostly enriched in mitosis-linked genes (*Figure 1D*), in agreement with the known cell proliferation arrest observed upon H2A.Z.1 depletion. Genes induced upon H2A.Z.2 knock-down are also enriched in cell cycle-linked genes (*Figure 1D*), although no obvious effect on cell proliferation could be observed upon H2A.Z.2 depletion (data not shown).

## H2A.Z.1 and H2A.Z.2 isoforms regulate both distinct and overlapping sets of genes

We next analysed whether H2A.Z.1 and H2A.Z.2 regulate the same set of genes. We found that among the 3573 genes regulated by H2A.Z.1, 759 are also regulated by H2A.Z.2 (Fig.ure 2A), whereas the expected overlap for random lists of gene of the same size would be 255. Actually, more than half of the genes regulated by H2A.Z.2 are also regulated by H2A.Z.1. This intersection is highly significant (p value = $9,38E^{-283}$), indicating that H2A.Z isoforms regulate overlapping sets of genes. However, they also have independent functions, since 2814 and 741 genes are regulated specifically by H2A.Z.1 and H2A.Z.2 respectively. The lists of genes up-regulated or down-regulated upon H2A.Z.1 or H2A.Z.2 depletion in WI38 cells are shown in *Supplementary files 1–4*.

We next crossed the lists of genes regulated by each H2A.Z isoform considering whether the genes were activated or repressed. We found 325 genes activated and 192 repressed by both H2AZ.1 and H2A.Z.2, both overlaps being highly significant (*Figure 2B*). Taking into account the 691 genes which were found deregulated only when we depleted the two isoforms, H2A.Z isoforms regulate 1208 genes in a similar way. This result indicates that both H2A.Z isoforms can play similar roles in the regulation of specific genes. However, we also found 72 genes repressed by H2A.Z.1 and activated by H2A.Z.2 and 170 activated by H2A.Z.1 and repressed by H2A.Z.2 (*Figure 2B*). This latter overlap is much more than expected by chance and is also highly significant, indicating that H2A.Z.1 and H2A.Z.2 can also regulate gene expression in an opposite fashion for a significant proportion of genes.

Using RT-qPCR, we analysed the expression of 6 selected mRNAs (ZDHHC20, RRM2, PLAT, COLEC12, AKAP12 and ADAMTS1) deregulated upon either H2A.Z isoform depletion. This showed a striking similarity with RNA-Seq results, validating the analysis (*Figure 2—figure supplement 1*). Moreover, results concerning the effects of H2A.Z.2 were confirmed with a second independent siRNA (*Figure 2—figure supplement 2*), ruling out the possibility of off-target effects at least for H2A.Z.2. Note however that we have not been able to find another efficient and specific siRNA against H2A.Z.1 (they were either inefficient or also decreased H2A.Z.2 levels). Some individual genes we identified here as regulated by H2A.Z.1 could thus be due to off-target effects. However, the highly significant intersection between genes differentially-expressed upon H2A.Z.1 and H2A.Z.2 depletion that we observed in WI38 as well as following experiments (see below) strongly suggests that off-target effects are modest. In particular, data on the ZDHHC20, RRM2 and PLAT gene were reproduced using a siRNA targeting a H2A.Z.1 interactor (see below), ruling out the possibility of off-target effects at least for these genes.

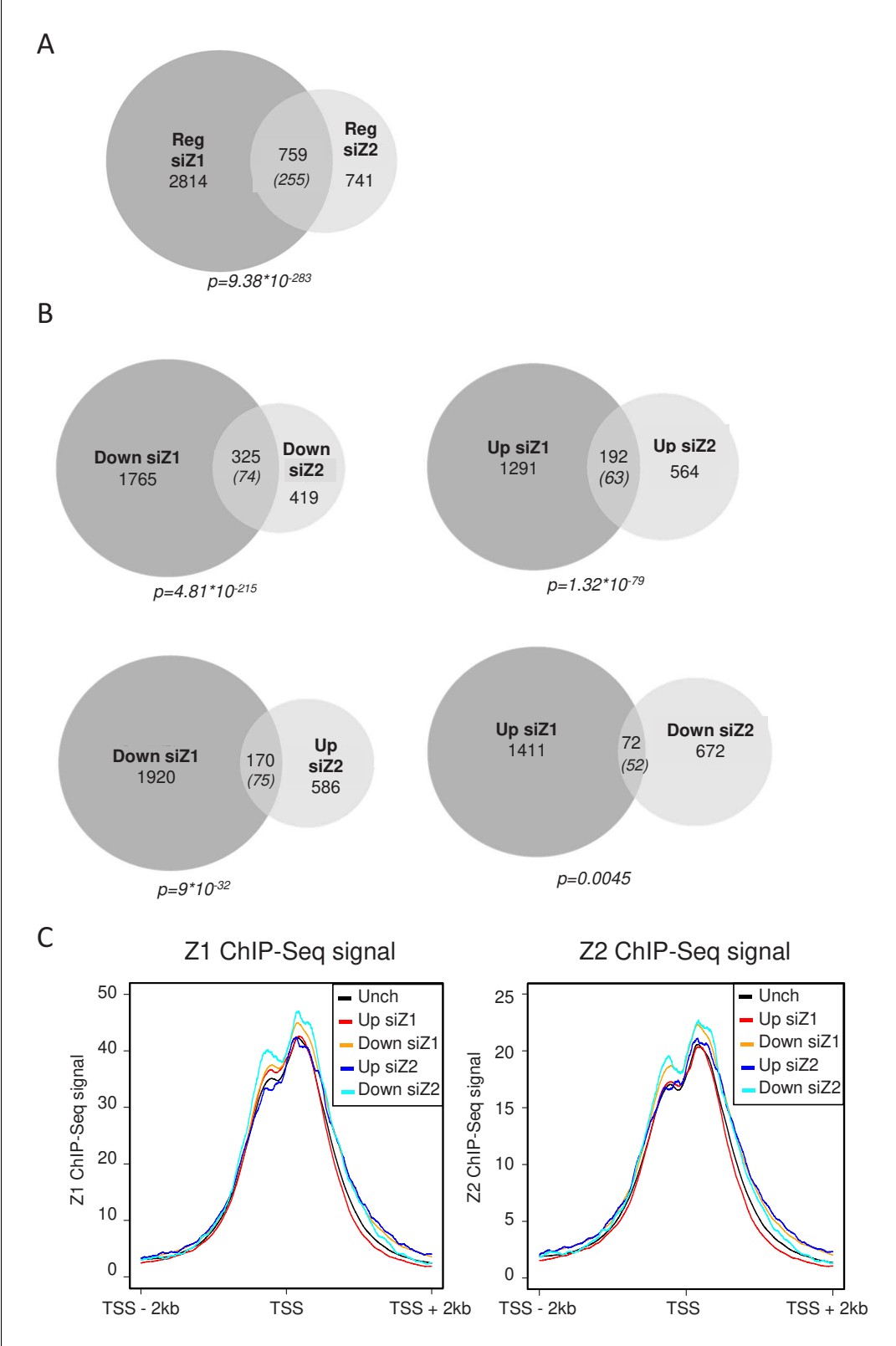

**Figure 2.** H2A.Z.1 and H2A.Z.2 regulate both distinct and overlapping sets of genes. RNA-Seq data was analysed for differential gene expression in samples transfected with either H2A.Z.1 siRNA or H2A.Z.2 siRNA versus the control siRNA sample. (**A**) Venn diagram showing the intersection between genes differentially expressed upon H2A.Z.1 and H2A.Z.2 inhibition. The p value indicated below the diagram indicates the significance of the intersection calculated using the Chi square test considering all expressed genes. The numbers in brackets indicate the expected number of genes

*Figure 2 continued on next page*

*Figure 2 continued*

considering the total number of expressed genes if intersection was random. (**B**) Same as in A, except that the intersections indicate genes that are up-regulated or down-regulated in each sample and those that are regulated in an opposite way. Note that the intersection between genes up-regulated upon H2A.Z.1 depletion and down regulated upon H2A.Z.2 depletion is not highly significant. (**C**) U2OS cells expressing endogenously Flag-tagged H2A.Z.1 or H2A.Z.2 were subjected to ChIP-Seq experiments using anti-Flag antibodies. Metadata showing ChIP-Seq signals around TSS were computed for the five classes of genes (Unch: unchanged upon H2A.Z.1 or H2A.Z.2 depletion) defined from RNA-Seq data obtained in U2OS upon H2A.Z.1 or H2A.Z.2 depletion (see *Figure 2—figure supplement 3*). A representative experiment is shown. A replicate is shown in Fig. *Figure 2—figure supplement 5A*.

The online version of this article includes the following source data and figure supplement(s) for figure 2:

**Figure supplement 1.** Validation of RNA Seq results.

**Figure supplement 1—source data 1.** Source data of the histogrammes representing the validation of RNA-seq effects by RT-qPCR on *Figure 2—figure supplement 1*.

**Figure supplement 2.** Effect of a second siRNA H2A.Z.2 (siZ2#).

**Figure supplement 2—source data 1.** Source Data for the histogrammes representing the effect of a second siRNA H2A.Z.2 (siH2A.Z.2#) on *Figure 2—figure supplement 2*.

**Figure supplement 3.** RNA-seq analysis after H2A.Z.1 and H2A.Z.2 depletion in U2OS cells.

**Figure supplement 3—source data 1.** Source data of the histogramme representing the depletion of H2A.Z.1 and H2A.Z.2 in response to siRNAs in U2OS cells in *Figure 2—figure supplement 3*.

**Figure supplement 4.** Profiles of tagged H2A.

**Figure supplement 5.** Analysis of H2A.Z.1 or H2A.Z.2 presence around TSS and enhancers.

Importantly, very similar results were observed in RNA-Seq data obtained following depletion of H2A.Z.1 or H2A.Z.2 in tumoral U2OS cells. We found more genes regulated by H2A.Z.1 (5196), less by H2A.Z.2 (673) than in WI38 cells, with roughly an equivalent number of activated and repressed genes (*Figure 2—figure supplement 3B*). The lists of genes up-regulated or down-regulated upon either H2A.Z.1, H2A.Z.2 or both depletion in U2OS cells are shown in *Supplementary files 6–10*. Again, crossing the results obtained for H2A.Z.1 and H2A.Z.2 indicated that, besides genes regulated by only one isoform, significantly enriched gene populations are activated or repressed by both H2A.Z.1 and H2A.Z.2, or activated by H2A.Z.1 and repressed by H2A.Z.2 (*Figure 2—figure supplement 3B*), confirming that H2A.Z.1 and H2A.Z.2 can regulate specific gene expression similarly or in an opposite way.

Finally, we crossed the results obtained in U2OS cells with those obtained in WI38 cells (*Figure 2—figure supplement 3C*). Despite significant overlap, these lists were mostly different, indicating that H2A.Z.1 and H2A.Z.2 regulate different sets of genes in different cells, consistent with the fact that transcription regulation by H2A.Z isoforms H2A.Z.1 and H2A.Z.2 is highly specific of the promoter context.

## The presence of H2A.Z isoforms at TSS does not determine their transcriptional effect

We next tested whether genes specifically regulated by a given isoform were characterized by a specific feature with respect to the presence of this isoform at their promoters. Thanks to the U2OS cells lines with tagged endogenous isoforms, we performed H2A.Z.1 and H2A.Z.2 ChIP-Seq experiments. ChIP-Seq profiles on the *CDKN1A/p21*, *GAPDH* genes were very similar and showed accumulation of H2A.Z.1 and H2A.Z.2 signal mostly at the TSS (*Figure 2—figure supplement 4*). We then integrated these data with the RNA-Seq following depletion of H2A.Z.1 or H2A.Z.2 in the same cells. Metadata analyses showed that both isoforms accumulate around the Transcription Start Sites (TSS) of expressed genes at the −1 and +one nucleosome surrounding the Nucleosome Depleted region (*Figure 2C*), as already shown for total H2A.Z (*Barski et al., 2007*). Importantly, binding of H2A.Z.1 appeared to be similar on genes that were unchanged upon depletion of H2A.Z.1 or H2A.Z.2 than on genes which responded to H2A.Z.1 depletion, either positively or negatively (*Figure 2C* and *Figure 2—figure supplement 5A*). Although genes which are repressed by H2A.Z.1 depletion showed a slightly higher level of H2A.Z.1, they also showed a higher level of H2A.Z.2 (*Figure 2C* and *Figure 2—figure supplement 5A*). Similarly, genes responding to H2A.Z.2 depletion do not show major differences in the amount of H2A.Z.2 bound to their promoters. These data thus indicate that genes affected by the depletion of a given isoform are not characterized by the amount of this

isoform around the TSS. A recent report described a higher H2A.Z.2/H2A.Z.1 ratio at active enhancers (*Greenberg et al., 2019*). To test whether this is also true in U2OS cells, we recovered U2OS enhancers through enhancer atlas (http://www.enhanceratlas.org/). We next computed the ratio of H2A.Z.1 to H2A.Z.2 ChIP-Seq signals at these enhancers as well as at all TSS or on control genomic regions. As previously found by *Greenberg et al. (2019)*, we found that the H2A.Z.1/H2A.Z.2 ratio was significantly lower at enhancers than at TSS (See *Figure 2—figure supplement 5B and C* for box plots and metadata). However, it was even lower at regions chosen arbitrary along the genomes (*Figure 2—figure supplement 5B and C*). To rule out any effect due to differences in the total amount of H2A.Z, we calculated this total amount by adding the H2A.Z.1 and H2A.Z.2 signal (which is feasible following internal spike-in normalisation) and sorted enhancers and promoters according to this amount. Strikingly, we did not observe a lower H2A.Z.1/H2A.Z.2 ratio at enhancers compared to promoters, whichever the class we considered (see *Figure 2—figure supplement 5D and E* for box plots and metadata) for the class with the highest level of total H2A.Z). Thus, in U2OS cells, the ratio H2A.Z.2/H2A.Z.1 is dependent on the total amount of H2A.Z present but not on any functional differences between enhancers and promoters.

## H2A.Z.1 and H2A.Z.2 isoforms can have antagonistic roles

We next asked whether the depletion of both isoforms together in WI38 non-transformed cells could lead to cumulative effects, as could be guessed if the total amount of H2A.Z was functionally important. As mentioned, only upon transfection of both siRNAs could we achieve efficient inhibition of total H2A.Z expression (see *Figure 1C*). We thus tested whether depletion of H2A.Z.2 could amplify the effects observed upon depletion of H2A.Z.1 alone. For the genes up-regulated upon H2A.Z.1 depletion, we plotted the effect of H2A.Z.1 depletion alone for each gene (calculating Log2 (siH2A.Z.1/siCtrl) for each gene) and the effect of both H2A.Z.1 and H2A.Z.2 depletion (Log2 (siH2A.Z.1+.2)/siCtrl)). Strikingly, we observed no cumulative effect of H2A.Z.1 and H2A.Z.2 depletion (*Figure 3A*). On the contrary, the effect of the double depletion was significantly lower than the effect of depleting H2A.Z.1 alone (*Figure 3A*). Importantly, similar findings were observed when analysing genes down-regulated upon H2A.Z.1 depletion (*Figure 3A*) and genes up- or down-regulated upon H2A.Z.2 depletion (*Figure 3B*). Thus, on these genes, the transcriptional effect of depleting one isoform on gene expression is attenuated upon depletion of the other isoform. This is not due to a lower siRNA effect upon co-transfection as shown in *Figure 1*. These results suggest that the effect of the loss of one H2AZ isoform on gene expression depends on the presence of the other H2AZ isoform. Thus, altogether these data uncover an antagonistic function of both H2A.Z isoforms on specific gene expression. Importantly, these findings were confirmed by RT-qPCR for the PLAT, AKAP12, ADAMTS1 and COLEC12 mRNAs (see *Figure 2—figure supplement 1*).

Note however that cumulative effects can be observed on genes which are deregulated only upon the combined depletion of the isoforms, as expected, as well as on some genes which are similarly regulated by H2A.Z.1 and H2A.Z.2 (*Figure 3—figure supplement 1*).

Again, very similar results were observed in U2OS cells, in which the effects of depleting one isoform were attenuated upon depletion of the other (*Figure 3—figure supplement 2*), with the notable exception of genes down-regulated upon H2A.Z.2 depletion.

Altogether, these data indicate a conserved complex interplay between H2A.Z.1 and H2A.Z.2 in specific gene regulation, with two types of H2A.Z-regulated genes: genes that they regulate similarly or on which they can compensate each other and genes that they differentially regulate and on which there is a rather general antagonism between the two isoforms.

## H2A.Z isoforms can replace each other at promoters

We next investigated the molecular mechanism underlying this complex interplay. H2A.Z is proposed to regulate specific gene expression by binding around gene Transcription Start Sites (TSSs) (*Subramanian et al., 2015*). One possibility could be that H2A.Z.1 and H2A.Z.2 can compete with each other for binding to the same promoters, with one isoform replacing the depleted one. Indeed, this would result in compensatory mechanisms where they play similar roles or in an antagonism where they bring about a different consequence regarding gene expression.

In agreement with this hypothesis, analysis of ChIP-Seq profiles (*Figure 2—figure supplement 5*) and ChIP-qPCR experiments (*Figure 4A*) using Flag antibodies indicate that both isoforms are

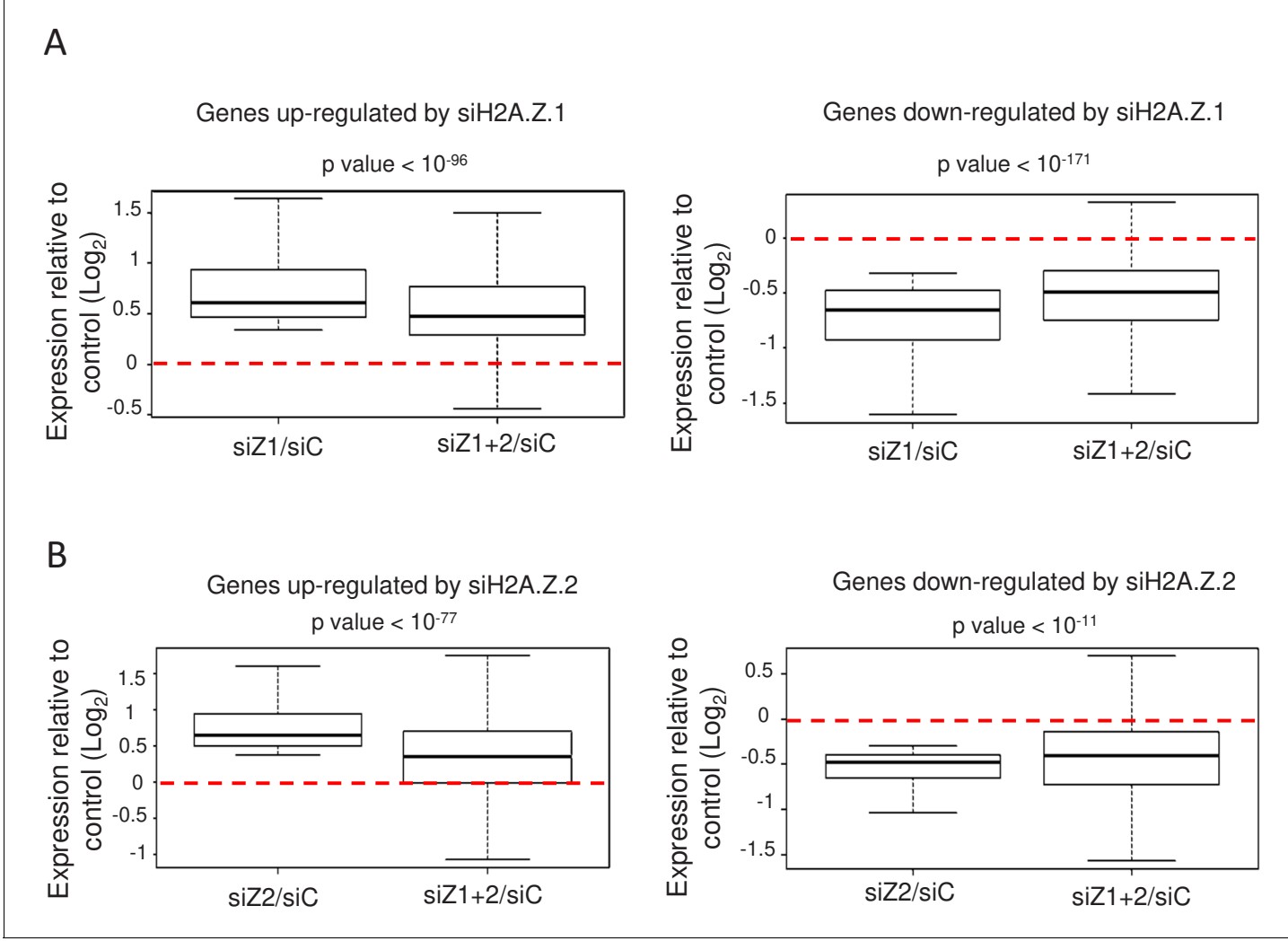

**Figure 3.** H2A.Z isoforms exert an antagonistic regulation on gene expression. (A) For each gene up-regulated (left) or down-regulated (right) upon H2A.Z.1 depletion, we calculated the ratio between its expression in either H2A.Z.1 depleted cells (siZ1/siC) or cells depleted for H2A.Z.1 and H2A.Z.2 versus control cells (siZ1+2/SIC). The boxplots show the median, the 25% percentiles and the extrema of the Log2 of this ratio within the gene population (without outliers). The p value shows the significance of the difference between the two populations (paired welch test). (B) Same as in A, except that the calculation was done for genes up-regulated (left) or down-regulated (right) upon H2A.Z.2 depletion.
The online version of this article includes the following figure supplement(s) for figure 3:

**Figure supplement 1.** Effect of the double depletion of H2A.Z.1 and H2A.Z.2 on genes similarly regulated by H2A.Z.1 and H2A.Z.2.
**Figure supplement 2.** Antagonistic regulation by H2A.Z isoforms in U2OS cells.

recruited to the *GAPDH* and *CDKN1A/p21* promoters in U2OS cells expressing endogenous H2A.Z with a 3xFlag-2xStrep tag. Moreover, at the genome-wide level, we observed a strong correlation between the levels of H2A.Z.1 and H2A.Z.2 binding to gene promoters (*Figure 4B*). Thus, these data indicate that H2A.Z.1 and H2A.Z.2 bind to the same genomic regions. We next tested whether depletion of one isoform could result in its replacement by the other. We transfected U2OS cells expressing 3xFlag-2xStrep-tagged H2A.Z.2 with siRNA against H2A.Z.1 and analysed chromatin recruitment of tagged H2A.Z.2 by performing a ChIP assay with Flag antibodies. Spike-in DNA was added to increase accuracy of the ChIP results. We found that H2A.Z.2 levels strongly increased upon H2A.Z.1 depletion at two loci at which H2A.Z.1 was bound, that is the *CDKN1A/p21* and *GAPDH* promoters, underlining the replacement of H2A.Z.1 by H2A.Z.2 upon H2A.Z.1 depletion (*Figure 4C*). In reciprocal experiments, we did not observe a significant increase in Flag-tagged H2A.Z.1 occupancy following depletion of H2A.Z.2 (*Figure 4D*), most likely because H2A.Z.2 is less

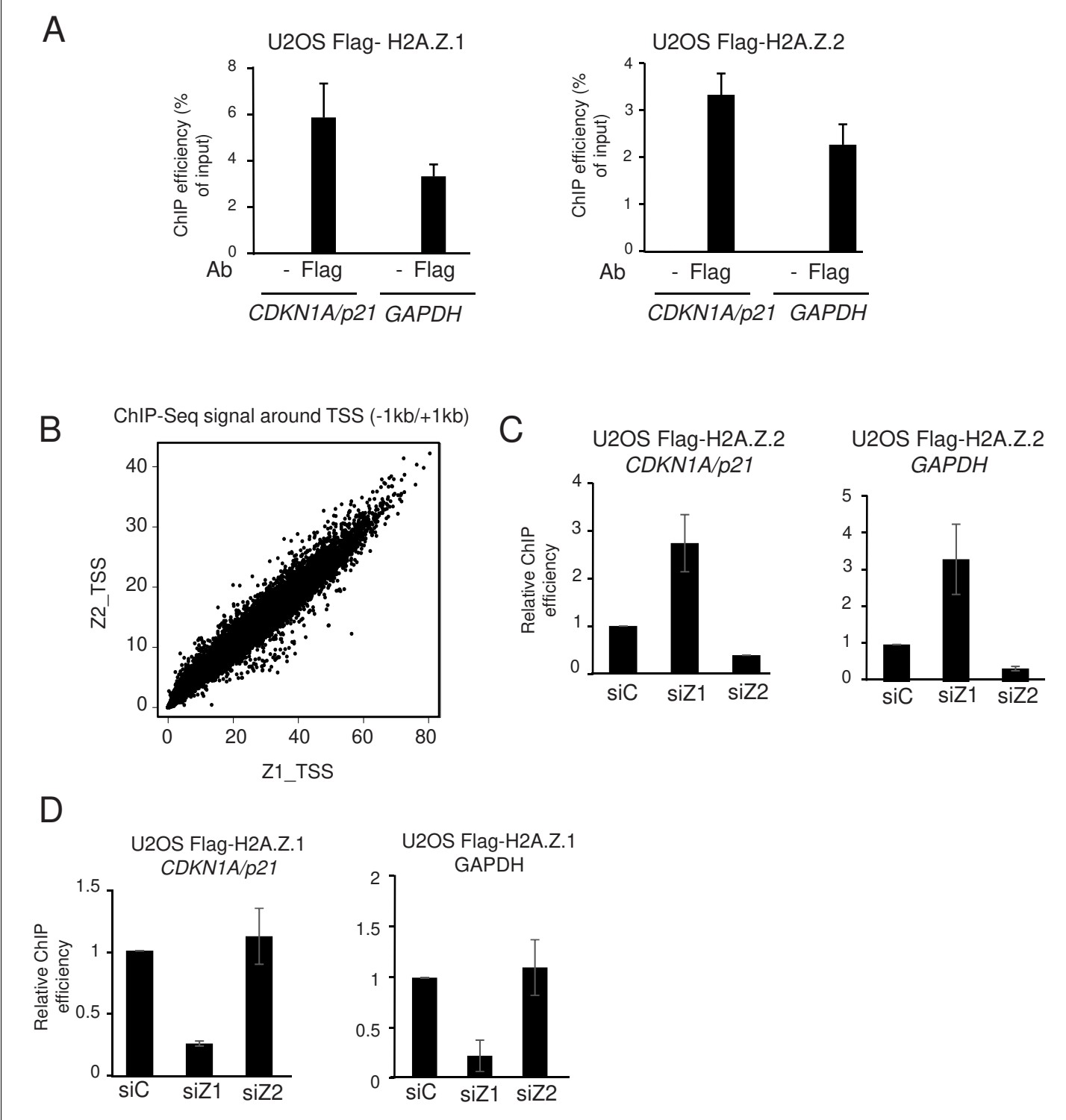

**Figure 4.** H2A.Z.1 and H2A.Z.2 replace each other at genes promoters. (**A**) U2OS cells expressing endogenously tagged H2A.Z.1 or H2A.Z.2 as indicated were subjected to a ChIP assay using the Flag antibody or no antibody as a control. The amount of the indicated sequences was measured by qPCR and calculated relative to the input DNA. The mean and SDOM from three independent experiments are shown. (**B**) A ChIP Seq assay was performed on the same samples. The amount of H2A.Z.1 or H2A.Z.2 from −1000 to +1000 of each protein-coding gene TSS was calculated and plotted against one another. Note the striking correlation between the binding of H2A.Z.1 and H2A.Z.2. (**C**) U2OS cells expressing endogenously tagged H2A.Z.2 were transfected with the indicated siRNA. 72 hr later cells were harvested and subjected to a ChIP experiment in the presence of spike-in chromatin. The amount of the indicated promoter was measured by qPCR, calculated relative to spike-in signals and relative to one for cells transfected

*Figure 4 continued on next page*

Figure 4 continued

with the control siRNA. The mean and SDOM from three independent experiments are shown. (D) Same as in C, except that U2OS cells expressing endogenously tagged H2A.Z.1 were used.

The online version of this article includes the following source data for figure 4:

**Source data 1.** Source Data of the histogrammes representing ChIP experiments in U2OS cells expressing tagged H2A.

**Source data 2.** Source Data of ChIP showing the competition between the two isoforms on *Figure 4C and D*.

expressed than H2A.Z.1 so that replacement of a significant amount of H2A.Z.2 by H2A.Z.1 does not increase much the total amount of H2A.Z.1. Nevertheless, these data suggest that H2A.Z isoforms can replace each other at TSS, therefore explaining the complex interplay we observed between H2A.Z isoforms for specific gene regulation.

At promoters at which they play a similar role, this could result in compensation for specific gene regulation, and deregulation would be seen only when depleted both isoforms. We observe this situation for 691 genes in WI38 and 800 genes in U2OS cells. At promoters at which H2A.Z.1 and H2A.Z.2 play a differential role, they would compete with each other, and that would result in the general antagonism we observe on the genes differentially regulated upon depletion of one isoform.

## H2A.Z.1 and H2A.Z.2 are differently associated with specific proteins

We then investigated the mechanism by which the presence of H2A.Z.1 or H2A.Z.2 could have differential functional consequences for gene transcription, despite their strong similarities. We thus asked whether they could be associated with different proteins and whether this could lead to different outcomes regarding gene expression. We tagged either isoform with a 3xFlag-2xStrep tag through genome editing in K562 cells (see the characterisation of both cell lines in *Figure 5—figure supplement 1*; *Dalvai et al., 2015*). Indeed, since these cells are non-adherent cells, they are much more convenient to grow in amounts large enough to purify and identify by mass spectrometry interactors of endogenous proteins. Flag western blots on these clones indicate that, like in U2OS cells, a band of higher molecular weight can be observed with both isoforms *Figure 5—figure supplement 1*, most likely representing ubiquitinated H2A.Z considering its size and the fact that it represents 20% to 25% of non ubiquitinated H2A.Z when extraction is performed in the presence of DUB inhibitors (data not shown). ChIP-Seq analysis indicate that in this cell line as in U2OS cells, the two isoforms localise at the TSS of active genes and we observed a strong correlation between levels of H2A.Z.1 and H2A.Z.2 around TSS (*Figure 5—figure supplement 1*). We next performed mass spectrometry analysis of proteins interacting with each isoform expressed at endogenous levels. Since endogenous H2A.Z.2 is less expressed than H2A.Z.1, we used clones heterozygously-tagged H2A.Z.1 vs homozygously-tagged H2A.Z.2 to adequately compare interactomes with similar expression levels of the bait (*Figure 5A* and *Figure 5—figure supplement 1*, *Supplementary file 8*). We found that the tandem affinity purification of each isoform from soluble nuclear extracts led to the co-fractionation of the previously characterized H2A.Z-incorporating complexes, that is p400- and SRCAP-containing complexes, and H2A.Z/H2B histone chaperones (see protein gel, mass spectrometry and western blot analysis in *Figure 5B–D* and *Figure 5—figure supplement 2*), in agreement with our findings that both isoforms can be incorporated at the same locations. Strikingly though, some proteins are found at higher levels in H2A.Z.1 purifications, such as chromatin proteins PHF14, HMG20A/iBRAF, TCF20 and RAI1, which were already found co-purifying together (*Eberl et al., 2013*), whereas others, such as SIRT1 are found at higher levels in H2A.Z.2 purifications (*Figure 5C–D* and *Figure 5—figure supplement 2A*). Importantly, similar preferential binding of PHF14 and SIRT1 to H2A.Z.1 and H2A.Z.2 respectively was also observed in U2OS cells (*Figure 5—figure supplement 2B*).

## PHF14 and SIRT1 are major H2A.Z1 and H2A.Z.2 effectors

We next tested whether the proteins interacting specifically with H2A.Z.1 and H2A.Z.2 could be recruited to chromatin through their interaction with H2A.Z.1 and H2A.Z.2. We could not obtain specific ChIP signals with commercially-available antibodies against PHF14. We thus raised a stable U2OS cell line in which one PHF14 allele was edited in order to express a flag tagged PHF14 protein

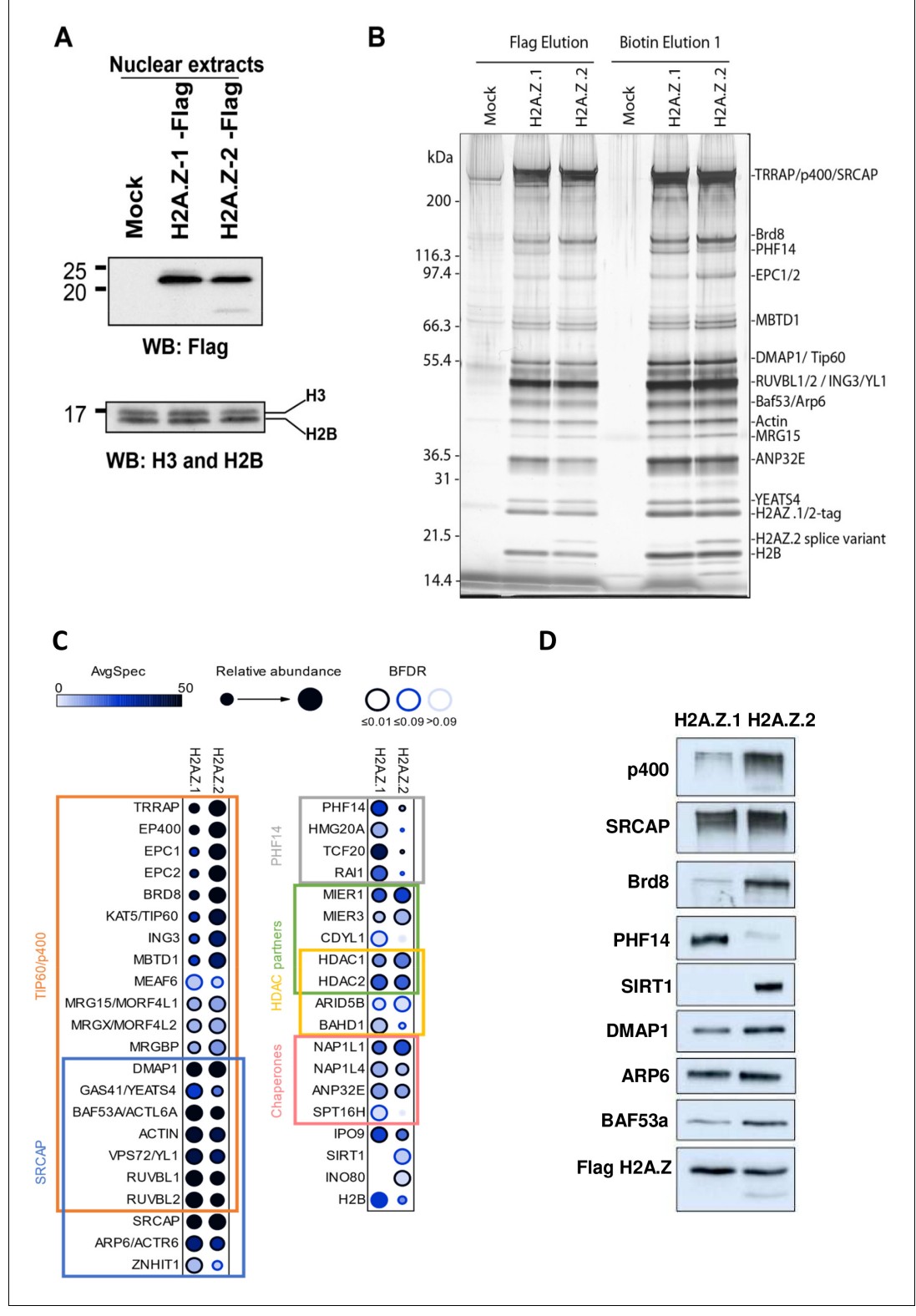

**Figure 5.** Identification of differential H2AZ.1 and H2AZ.2 interactors. (**A**) Comparison of expression levels of tagged endogenous H2A.Z.1 (heterozygous) and H2A.Z.2 (homozygous) clones used in tandem affinity purification from nuclear extracts (see *Figure 5—figure supplement 1*). (**B**) Silver stained gel of fractions obtained for the purification of H2A.Z.1 and H2A.Z.2 from nuclear extracts of K562 cells shown in (**A**). A mock non-tagged cell line is used as control. Flag peptide elution is obtained from the first purification step (M2-Flag resin) and biotin elution
*Figure 5 continued on next page*

*Figure 5 continued*

from the second final step (Strep-Tactin resin). Known components of protein complexes interacting with histone H2A.Z are identified on the right (PHF14 is also indicated). (C) Dot-blot representation of AP-MS experiments using tagged H2A.Z.1 and H2A.Z.2 as baits. Circle filling represents average spectral counts, while circle diameter represents relative enrichment in one bait versus the other and circle border represents BFDR confidence level. Known/expected partners based on the literature and large-scale public data (BioGrid) include TIP60/p400, SRCAP and HDAC complexes. Data represent two replicates for each bait and were normalized on H2AZ-H2B chaperone levels (ANP32E, NAP1L1 and NAP1L4). (D) Western-blot validation of interactions shown in (B–C). TAP-purified fractions were normalized based on Flag-H2A.Z signals, loaded on SDS-PAGE gels, and blotted with the indicated antibodies.

The online version of this article includes the following figure supplement(s) for figure 5:

**Figure supplement 1.** Tagging of H2A.Z isoforms by CRISPR/Cas9 in K562 cells used for characterisation of interactomes.

**Figure supplement 2.** Mass spectrometry analysis of the H2A.Z.1/2 purifications shown in *Figure 5B* and validation in U2OS cells.

(see *Figure 6—figure supplement 1* for the characterisation of this cell line). Despite epitope tagging, PHF14 ChIP experiments did not give any specific signal on the genes we tested (data not shown). To understand the molecular interplay of SIRT1 and PHF14 with H2A.Z isoforms, we performed fractionation experiments using the U2OS flag-tagged PHF14 cell line. We first began by a cell fractionation procedure relying on EDTA-mediated bivalent ions chelation. In these conditions, SIRT1 is mostly in the nuclear and cytoplasmic fractions, with detectable amount in the chromatin fraction (*Figure 6A*). Interestingly, depletion of H2A.Z.2 resulted in a decrease in SIRT1 presence in the chromatin fraction (*Figure 6A*, see *Figure 6—figure supplement 2* for the quantification and a replicate), indicating that H2A.Z.2 favours the chromatin localisation of SIRT1.

In these conditions, PHF14 was found exclusively at the chromatin (*Figure 6A*), and this irrespective of the siRNA used (data not shown). We thus tested fractionation conditions in which we extract proteins from chromatin using high salt conditions (420 mM NaCl). In these conditions, we observed that PHF14 is present both in the chromatin fraction and in the soluble nuclear fraction (*Figure 6B*). Depletion of H2A.Z.1 (see the H2A.Z western blot shown in *Figure 6—figure supplement 1*) does not lead to a reproducible decrease of PHF14 in the chromatin fraction (*Figure 6B* and *Figure 6—figure supplement 2* for the quantification and a replicate). However, we consistently observed an increase of PHF14 amounts in the soluble nuclear fraction. Since there is no PHF14 in the cytoplasmic fraction in any of the conditions tested, this fractionation procedure resulted in higher PHF14 extraction from chromatin upon H2A.Z.1 knockdown, suggesting that H2A.Z.1 favours the interaction of PHF14 with chromatin.

Taken together, these results are consistent with the hypothesis that H2A.Z.1 and H2A.Z.2-containing nucleosomes provide docking sites in chromatin for PHF14 and SIRT1 respectively.

To investigate whether PHF14 and SIRT1 could be responsible for the effect of H2A.Z.1 and H2A.Z.2 on gene expression, we analysed whether PHF14 and SIRT1 depletion could phenocopy H2A.Z isoform depletion. To do so, we transfected immortalized WI38 human primary fibroblasts with siRNAs against SIRT1 or PHF14. These siRNAs were efficient as verified at the mRNA level by RT-qPCR and at the protein level by western blot analysis (*Figure 6—figure supplement 3*). Furthermore they had no effect on total H2A.Z or H2A.Z.1 or H2A.Z.2 mRNA expression levels (*Figure 6—figure supplement 3*). Analysis of specific gene expression by qPCR indicated that on four out of the five genes we analysed, PHF14 depletion induces changes in a similar way compared to H2A.Z.1 depletion (*Figure 6C*, compared to *Figure 2—figure supplement 1*). Similar results were obtained when comparing the effects of SIRT1 depletion with H2A.Z.2 depletion, with three out of the five analysed genes regulated in a similar manner (*Figure 6C*). Note however that fold changes on these genes upon depletion of H2A.Z isoforms or their effectors can be very different, indicating that other mechanisms may take place.

To confirm this finding at the genome wide level, we performed RNA-Seq analysis following depletion of SIRT1 and PHF14. We found 4189 and 2405 genes significantly regulated by PHF14 and SIRT1 respectively (see *Supplementary files 11* and *12* for the complete list of de-regulated genes upon PHF14 and SIRT1 depletion, respectively). Strikingly, we observed that 30.6% of up-

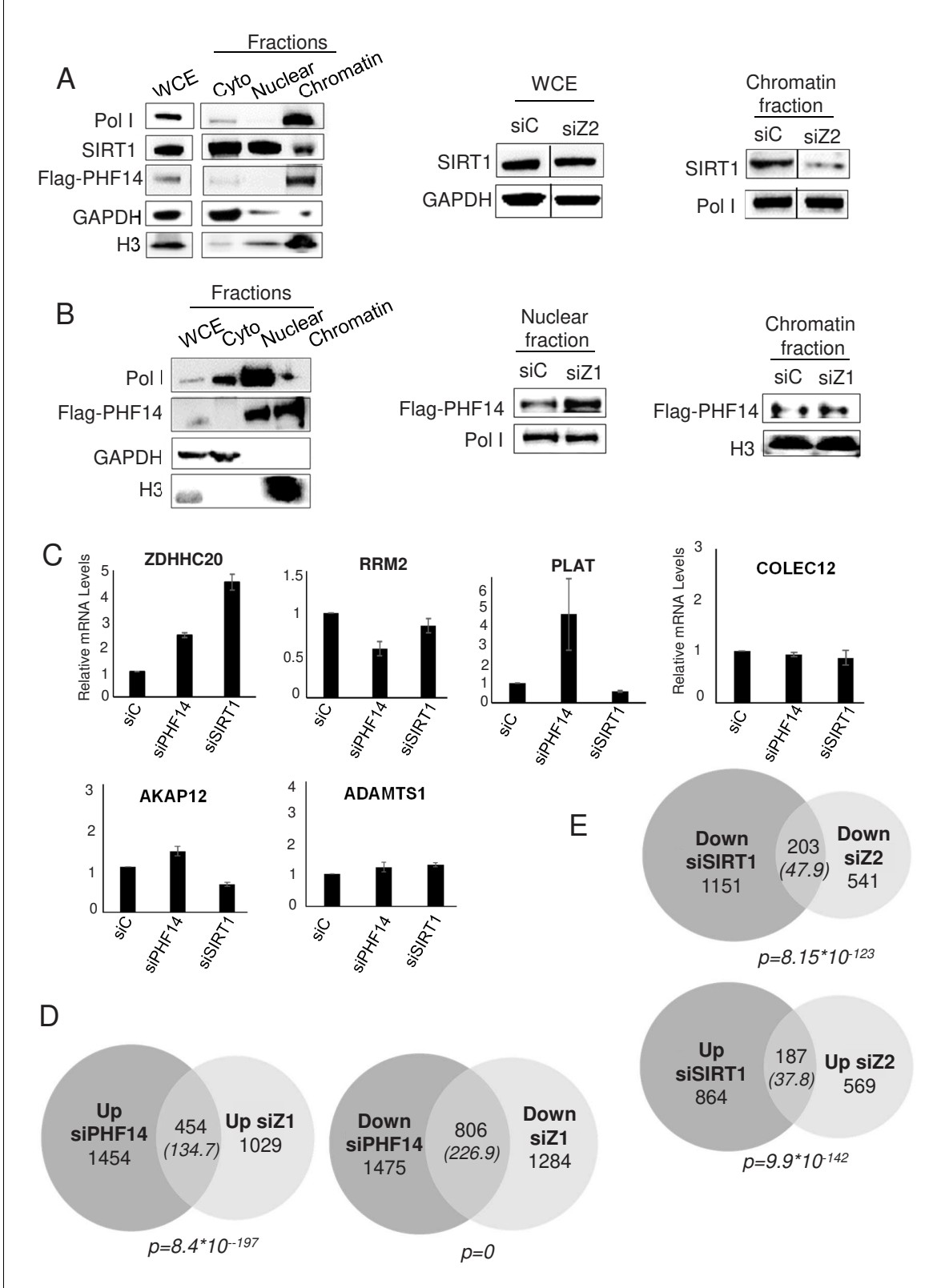

**Figure 6.** PHF14 and SIRT1 mediate H2A.Z.1 and H2A.Z.2 functions respectively. (**A**) Left panel: U2OS cells expressing endogenously tagged PHF14 were subjected to cell fractionation experiments in mild conditions. Cell fractions were then subjected to western blot analysis using the indicated antibody. Right panel: U2OS cells expressing endogenously tagged PHF14 were transfected with the indicated siRNA and analysed 72 hr later as in the left panel. A representative experiment from three independent experiments are shown (see *Figure 6—figure supplement 2* for a replicate) B) Left

*Figure 6 continued on next page*

Figure 6 continued

panel: U2OS cells expressing endogenously tagged PHF14 were subjected to cell fractionation experiments using NP40. Cell fractions were then subjected to western blot analysis using the indicated antibody. Right panel: U2OS cells expressing endogenously tagged PHF14 were transfected with the indicated siRNA and analysed 72 hr later as in the left panel. A representative experiment from three independent experiments are shown (see *Figure 6—figure supplement 2* for a replicate) C) WI38 cells were transfected with the indicated siRNA. 72 hr later, total RNA was prepared and analysed by RT-qPCR to assess the expression of the indicated genes. Data was standardised using GAPDH mRNA levels and calculated relative to one for cells transfected with the control siRNA. The mean and SDOM from three independent experiments are shown. (**D and E**) WI38 cells were transfected with the indicated siRNA. 72 hr later, total RNA was prepared and further purified to be subjected to RNA-Seq. Genes differentially expressed upon PHF14 or SIRT1 depletion were characterised. The Venn diagrams indicating the intersection of genes regulated by PHF14 and H2A.Z.1 (**D**) or by SIRT1 and H2A.Z.2 (**E**) are shown. The p value indicates the significance of the intersection (Chi square test) considering the total number of expressed genes. The numbers in brackets indicate the expected number of genes considering the total number of expressed genes if intersection was random.

The online version of this article includes the following source data and figure supplement(s) for figure 6:

**Source data 1.** Source Data of histogrammes on *Figure 6C* representing the validation by RT-qPCR of the RNA-seq after siSIRT1 and siPHF14.
**Figure supplement 1.** Characterisation of U2OS cells expressing 3xFlag-2xStrep tagged PHF14.
**Figure supplement 2.** Quantification and replicate of *Figure 6A and B* experiments.
**Figure supplement 3.** Validation of siRNAs targeting SIRT1 and PHF14.
**Figure supplement 3—source data 1.** Source Data of histogrammes on *Figure 6—figure supplement 3A* representing the efficiency of siRNA against SIRT1 and PHF14.
**Figure supplement 3—source data 2.** Source Data of histogrammes on *Figure 6—figure supplement 3B* representing the effect of siSIRT1 and siPHF14 on H2A.Z.1 and H2A.Z.2 mRNAs.

regulated genes and 38.5% of down-regulated genes in the siH2A.Z.1 condition were similarly regulated when PHF14 was silenced, which is a highly significant intersection (p=8,4*10$^{-197}$and 0 respectively) (*Figure 6D*). Likewise, 24.7% of up-regulated genes and 27.2% of down regulated genes in the siH2A.Z.2 condition were de-regulated in a similar way in the siSIRT1 condition (*Figure 6E*).

Thus, taken together, these data indicate that PHF14 and SIRT1 are major mediators of H2A.Z.1- and H2A.Z.2-specific gene regulation, since i) depletion of H2A.Z isoforms specifically affects their localisation ii) they affect transcription of a significant proportion of H2A.Z-regulated genes in a manner similar to the H2A.Z isoforms. Given that these effects can be positive or negative, our data further underline the function of PHF14 and SIRT1 in mediating the context-dependent regulation of specific gene transcription by H2A.Z isoforms.

## PHF14 and SIRT1 can mediate the antagonistic relationship between H2A.Z.1 and H2A.Z.2

Strikingly, when we analysed the PLAT mRNA, we found that co-depletion of SIRT1 along with H2A.Z.1 or PHF14 also abolishes the effects of H2A.Z.1 and PHF14 depletion (*Figure 7A*). This shows that SIRT1 mediates the antagonistic effect of H2A.Z.1 and H2A.Z.2 on PLAT expression. Very similar results were observed for the AKAP12 mRNA concerning it specific repression by H2A.Z.2 (which is mimicked by depletion of SIRT1). Indeed, not only is this repression attenuated upon co-depletion of H2A.Z.1 along with H2A.Z.2 or SIRT1, as shown in *Figure 3*, but also upon co-depletion of PHF14 along with H2A.Z.2 or SIRT1 (*Figure 7A*). Consistently, H2A.Z.2-dependent activation of the ADAMTS1 and COLEC12 mRNAs was attenuated upon depletion of H2A.Z.1 or PHF14 (*Figure 7—figure supplement 1A*). Thus, these results indicate that PHF14 and SIRT1 can antagonise each other, at least for the four genes we tested.

We next investigated the mechanism involved in this antagonism. Given that SIRT1 is a protein deacetylase, we tested whether PHF14 could favour protein acetylation. Indeed, PHF14 and associated proteins were already proposed to function by counteracting the function of the repressive LSD1-CoREST complex, which contains histone deacetylases along the H3K4 demethylase (*Garay et al., 2016*; *Wynder et al., 2005*). Therefore, we depleted PHF14 using a specific siRNA and assessed whether it affected global site-specific histone acetylation by western blot analysis. Importantly, depletion of PHF14 strongly decreased global histone H3K9 acetylation, a known SIRT1-target sites (*Vaquero et al., 2004*), while it had only a mild effect, if any, on histone H3K14 acetylation levels (*Figure 7B*, see *Figure 7—figure supplement 1B* for a replicate) To test whether this global change in H3K9 acetylation could also be observed at promoters regulated by H2A.Z.1

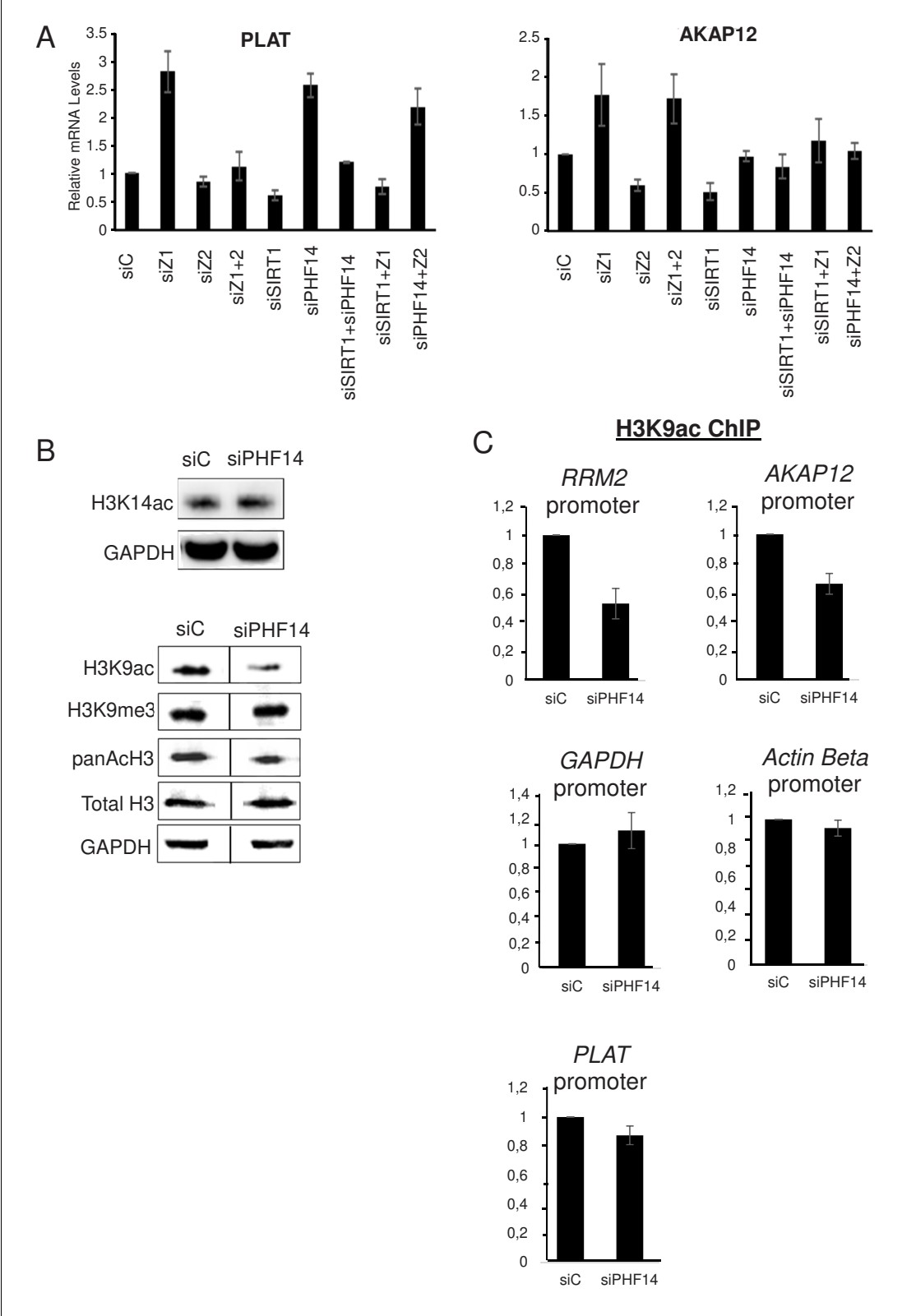

**Figure 7.** PHF14 and SIRT1 can mediate the antagonism between H2A.Z.1 and H2A.Z.2. (**A**) WI38 cells were transfected with the indicated siRNAs alone or in combination. 72 hr later, total RNA was prepared and analysed by RT-qPCR to assess the expression of the indicated genes. Data was standardised using GAPDH mRNA levels and calculated relative to one for cells transfected with the control siRNA. The mean and SDOM from three independent experiments are shown. (**B**) WI38 cells were transfected with the indicated siRNA. 72 hr later, whole cell extracts were prepared and

*Figure 7 continued on next page*

*Figure 7 continued*

subjected to western blot analysis using the indicated antibody. A replicate is shown in *Figure 7—figure supplement 1*. (**C**) WI38 cells were transfected with the indicated siRNA. 72 hr later, transfected cells were subjected to a ChIP assay using antibodies directed against H3K9ac (left) or anti-H3 (right).*Drosophila*or mouse spike-in chromatin was added after sonication as an internal control. The amount of the indicated human promoters was quantified by qPCR, standardised using the spike-in signal and calculated relative to one for control cells. The mean and SDOM from three independent experiments are shown. Legends to Supplementary Figures.

The online version of this article includes the following source data and figure supplement(s) for figure 7:

**Source data 1.** Source Data of histogrammes in *Figure 7C* representing ChIP H3K9 after PHF14 depletion on different promoters.
**Figure supplement 1.** PHF14 mediates antagonistic effect of H2A.Z.1 and H2A.Z.2 on ADAMTS1 and COLEC12 mRNA expression.
**Figure supplement 1—source data 1.** Source Data of histogrammes on *Figure 7A* and on *Figure 7—figure supplement 1B* showing that the antagonism between H2A.Z.1 and H2A.Z.2 is mediated by SIRT1 and PHF14.

and H2A.Z.2, we performed a ChIP assay against this acetylation mark. Spike-in chromatin was added to the samples as an internal control to increase accuracy of the ChIP measurements. We found that PHF14 depletion led to a decrease in H3K9 acetylation at two promoters regulated by H2A.Z.1 and H2A.Z.2, the *RRM2* and *AKAP12* promoters (*Figure 7C*). Note that no major effect could be observed on two not targeted promoters (*GAPDH* and *β-actin*) and on the H2A.Z.1-regulated *PLAT* promoter Thus, altogether these data indicate that PHF14 favours histone acetylation, at least at the *RRM2* and *AKAP12* promoters.

## Discussion

Here, we performed an integrated study of the role of H2A.Z isoforms H2A.Z.1 and H2A.Z.2 on gene expression in non-transformed cells.

We first confirmed that H2A.Z is a major regulator of specific gene expression. If we take into accounts all genes differentially regulated upon depletion of H2A.Z.1, H2A.Z.2 or both together, we found that 5005 and 6317 genes are deregulated in WI38 and U2OS cells, which represent roughly 1/4th of all expressed genes. Interestingly, the number of genes activated by either H2A.Z isoform is roughly equivalent to the number of genes they repress, in agreement with previous studies underlining a positive and negative role for H2A.Z in gene expression (*Dunn et al., 2017*; *Subramanian et al., 2015*).

We also found that there are many genes at which H2A.Z isoforms play similar roles including some on which they can compensate each other. This compensation probably explains why a strong phenotype is observed upon depletion of both H2A.Z isoforms in mouse intestinal stem cells whereas depletion of individual isoforms had no or weak effects (*Zhao et al., 2019*). At these promoters, the presence of H2A.Z could regulate gene expression by affecting nucleosome structure or stability, or by recruiting proteins leading to similar outcomes on gene expression.

We further found that both isoforms regulate different sets of genes, as previously shown (*Dunn et al., 2017*; *Greenberg et al., 2019*), again confirming that despite their high similarity, they can play different roles. Finally, we found that on a significant number of genes, H2A.Z.1 and H2A.Z.2 can play opposite roles, with H2A.Z.1 acting as an activator and H2A.Z.2 as a repressor. Interestingly, at these two gene populations, analysis of the double depleted samples uncovers a general antagonism between H2A.Z.1 and H2A.Z.2.

Altogether, these data underline that H2A.Z isoforms H2A.Z.1 and H2A.Z.2 can regulate gene expression similarly and antagonistically depending on the gene, uncovering a complex interplay.

We also provide major insights into the mechanism of this interplay. By ChIP-Seq analysis, we found H2A.Z isoforms are present on most promoters of expressed genes or genes prone to be expressed, as already shown for total H2A.Z (*Barski et al., 2007*). Clearly, the presence of a H2A.Z isoform around a TSS is not sufficient to predict the effect of this isoform on gene transcription. Rather, our ChIP-Seq data indicate that H2A.Z.1 and H2A.Z.1 are incorporated at the same promoters and we found that they can replace each other. This replacement could explain the compensation between the two isoforms we observe for the regulation of many promoters.

These data also suggest that the two isoforms compete each other for the binding to gene promoters. As a consequence, at a promoter specifically regulated by a given isoform, the presence of the other isoform would counteract the presence and the function of the regulatory isoform. Such a

model would explain the rather general antagonism we observe on genes specifically regulated by an isoform. It underlines the functional importance of the relative levels between the two isoforms.

Thus, the same mechanism, that is the ability to be incorporated at the same promoters, could explain both the compensation between H2A.Z isoforms at some promoters and their antagonism at others. It also fits with the fact that they can both interact with incorporation machineries (p400- and SRCAP complexes).

We next investigated how H2A.Z isoforms could differentially regulate gene expression. We identified proteins interacting preferentially with one isoform or the other, that is PHF14 with H2A.Z.1 and SIRT1 with H2A.Z.2. To our knowledge, this is the first demonstration that endogenous H2A.Z.1 and H2A.Z.2 can differentially bind to proteins. Note however that it was previously found using overexpressed proteins that H2A.Z.1 nucleosomes were slightly more efficient than H2A.Z.2 nucleosomes in interacting with BRD2 as well as with PHF14, HMG20A and TCF20 (*Draker et al., 2012*; *Pünzeler et al., 2017*).

A functional interaction between H2A.Z and SIRT1 in the nucleus was also already described (*Baptista et al., 2013*; *Chen et al., 2006*). The authors propose that deacetylation of H2A.Z by SIRT1 leads to its destabilization. However, we did not find any effect of SIRT1 depletion on H2A.Z expression in our hands. Moreover, such a mechanism would not explain the similarities in gene expression control by H2A.Z.2 and SIRT1. We rather found that depletion of H2A.Z.2 decreased the nuclear retention of SIRT1, in agreement with the interpretation that the H2A.Z.2/Sirt1 interaction is required for recruitment of SIRT1 in the nucleus.

Functional analysis of PHF14 and SIRT1 indicates that they can mediate, at least in part, the specific effects of H2A.Z.1 and H2A.Z.2 on gene expression. In particular, we provide evidence that PHF14 and SIRT1 can mediate the functional antagonism between both H2A.Z isoforms. These two proteins would function as readers of the presence of a specific H2A.Z isoform at gene promoters. Upon depletion of one isoform, the other isoform takes over, resulting in changes in the amount of promoter-bound PHF14 and/or SIRT1. These changes may in turn affect gene expression in a context-dependent manner. Importantly, we found that PHF14 favours acetylation of histones, at least on H3K9 residues at its target promoters. Thus, this may provide a mechanism by which PHF14 and SIRT1 mediate opposite effects on gene expression: when H2A.Z.1 is depleted, there is less recruitment of PHF14 and more recruitment of SIRT1, shifting the balance towards local deacetylation, whereas when H2A.Z.2 is depleted, the balance is shifted towards local acetylation. Such changes in acetylation of histones or non-histone proteins could mediate the context-dependent regulation of gene expression. However, the fact that there are many genes regulated by H2A.Z isoforms H2A.Z.1 and H2A.Z.2 but not by PHF14 and SIRT1 suggests that other mechanisms mediate H2A.Z gene expression regulation, such as changes in nucleosome stability, as previously proposed, or interaction with other specific effectors.

Our data thus indicate that the relative levels of H2A.Z isoforms is probably a major determinant of gene regulation and that changes in these levels lead to important changes in gene expression. Interestingly, the two H2A.Z isoforms are produced by two genes with two independent promoters (*Matsuda et al., 2010*), providing a basis for the regulation of the relative isoforms levels by signalling pathways. In this respect, it will be of great interest to decipher the transcription factors and the signalling pathways regulating H2A.Z promoters. For example, we recently showed that the *H2AFZ* promoter is a direct target of the Wnt signalling pathway (*Rispal et al., 2019*). Furthermore, it is known that the expression of both isoforms is regulated in a tissue- and developmental stage-dependent manner (*Dryhurst et al., 2009*). Our data suggest that these changes are essential for establishing the correct gene expression pattern associated with specific developmental pathways.

In addition to regulation of the H2A.Z.1/H2A.Z.2 relative levels, the potential regulation of the association between PHF14, SIRT1 or other readers with H2A.Z isoforms is clearly worth investigating. Our work points to the importance of the three residues differing between the two isoforms in the binding of specific partners. Strikingly, one of these differences (T14 in H2A.Z.1, A14 in H2A.Z.2) is located in the N-terminal tail basic patch important for H2A.Z function and target of multiple post-translational modifications such as acetylation, methylation or ubiquitylation. Furthermore, T14 of H2A.Z.1 could itself be phosphorylated which would likely affect adjacent lysine modifications (K13/15). Acetylation of H2A.Z has indeed been linked to transcriptional activation by many studies (*Bruce et al., 2005*; *Millar et al., 2006*; *Valdes-Mora et al., 2012*). It is tempting to speculate that post-translational modifications of this T in H2A.Z.1 or surrounding amino acids in H2A.Z.1 or H2A.

Z.2 could regulate the recruitment of H2A.Z.1 and H2A.Z.2 effectors, among which PHF14 and SIRT1, and therefore participate in the regulation of specific gene expression. In agreement with this hypothesis, the PHF14-complex contains many different PhD fingers, suggesting that it could function as a reader of H2A.Z.1 post-translational modifications.

Interestingly, a recent study showed that a S38T substitution in H2A.Z.1, mimicking T38 found in H2A.Z.2, rescues the SRCAP-dependent Floating-Harbor Syndrome (*Greenberg et al., 2019*). Understanding the molecular determinants of PHF14 complex and SIRT1 binding to H2A.Z isoforms will be a first step towards the identification of the mechanisms underlying the context-dependent regulation of transcription by H2A.Z isoforms.

# Materials and methods

## Cell cultures and transfections

Lung Fibroblastic cells WI38 hTERT RAF1-ER, which are immortalized by hTERT expression and contain an inducible RAF1 oncogene fused to estrogen receptor (ER) (*Jeanblanc et al., 2012*), were grown in minimum essential medium (MEM) supplemented with 10% fetal bovine serum (FBS), sodium pyruvate, L-glutamine, non-essential amino acids and penicillin-streptomycin, in normoxic culture conditions (5%O2).

Osteosarcoma U2OS cells were obtained from the ATCC and were grown in DMEM media with Glutamax (1 g/L glucose), supplemented with 10% of FBS, sodium pyruvate and penicillin-streptomycin.

K562 cells were obtained from the ATCC and maintained at 37°C under 5% $CO_2$ in RPMI medium supplemented with 10% newborn calf serum (Wisent) and GlutaMAX. When cultivated in spinner flasks, 25 mM HEPES-NaOH (pH 7.4) was added. Cells were transfected using the Amaxa 4D-Nucleofector (Lonza) per the manufacturer's recommendations.

All cell lines were regularly checked for the absence of mycoplasma contamination.

siRNA transfection was performed using the Dharmafect four reagent (Dharmacon) according to the manufacturer's instructions. 24 hr before transfection, 1.4 million of Wi-38 cells, and 850,000 of U2OS cells were plated in 10 cm dish. 100 nM of siRNA were used, and an equal volume of the culture medium was added 24 hr after transfection. 48 hr later, cells were harvested. The list of siRNAs used is indicated in *Supplementary file 13*. DNA plasmids were transfected with jet-PEI reagent (polyplus) according to manufacturer's instructions.

## Tagging of endogenous proteins in K562 cells

Cytomegalovirus (CMV)-driven human-codon optimized Cas9 nuclease (Addgene #41815) was used. gRNA sequences targeting *H2AFZ* and *H2AFV* were selected using a web-based tool (https://www.benchling.com/, using algorithms from *Hsu et al., 2013*, and *Doench et al., 2016*) and validated by Surveyor assay (Integrated DNA Technologies). gRNA expression vectors were built in the MLM3636 (Addgene #43860) backbone. The donor plasmid for *H2AFZ* was synthesized as a gBlock gene fragment (Integrated DNA Technologies) and assembled using the Zero Blunt TOPO cloning kit (Invitrogen). The donor plasmid for *H2AFV* was obtained by cloning. Briefly, a PCR fragment of 1.7 kb genomic DNA (1008 bp before stop codon and 690 bp after stop codon) was integrated in Zero Blunt TOPO plasmid following manufacturer's instructions. Subsequently, the PAM motif corresponding to the gRNA was mutated and the 3xFlag-2xStrep sequence (*Dalvai et al., 2015*) was integrated before the stop codon using Gibson assembly kit (NEB, E5510).

The i53 53BP1 inhibitor (Addgene #74939, *Canny et al., 2018*, gift from Amélie Fradet-Turcotte) was used to increase tagging efficiency of *H2AFZ* by homology directed repair. One million K562 cells were transfected using 2 µg gRNA plasmid, 2 µg Cas9 vector, and 4 µg donor (plus 600 ng i53 plasmid for *H2AFZ*). Limiting dilution cloning was performed 3 days post-transfection, and targeted clones were identified via out-out PCR as before (*Dalvai et al., 2015*).

## Generation of genome-edited cell clones in U2OS cells

We used the ouabaine based co-selection strategy described by *Agudelo et al. (2017)*. This consist on the co-transfection of a RNA guide + DNA donor that are able to give the cell a resistance to a drug, ouabaine, by inducing a mutation in the ATP1A1 gene (Na+/K+ pump). For this, we cloned

our RNA guides of interest in a modified pX330 plasmid (Addgene #86616, ATP1A1 G3 dual sgRNA) containing in addition to the CRISPR cas9 enzyme, the RNA guide needed to mutate ATP1A1 gene.

RNA guides for targetting *H2AFZ* and *H2AFV* are described above. RNA guides for the Flag-PHF14 CRISPR were selected using the CRISPOR website (http://crispor.tefor.net/). Primers were then chosen according to the website proposition. To clone guides, a phosphorylation step consisting on incubation of 1 µl of each oligo (100 µM) at 37°C for 30 min in the presence of the PNK enzyme (Promega) was performed. Then, samples were heated for 5 min at 95°C, and cooled over night at room temperature. The annealed phosphorylated product was then digested-ligated in a one step reaction into the ATP1A1 G3 dual sgRNA plasmid in the presence of bbs1 restriction enzyme (NEB) and T7 ligase (NEB), and the reaction was subjected to the following PCR cycle (37°C 5 min, 25°C 5 min, 6 times). The product of the ligation was exonuclease-digested using exonuclease V RecBCD for 30 min at 37°C. Competent cells were then transformed with the final product, and positive clones were selected by bbs1 digestion. 200,000 U2OS cells plated in 6-well plates were transfected with this plasmid, a double stranded 3xFlag-2xStrept tagged PHF14 DNA donor (300 bp, ordered from GeneScript) or donor plasmids for H2AFZ and H2AFV described above and Ouabaine-resistance donor (ordered from Addgene #66551, ATP1A1 plasmid donor) and guide to a final DNA concentration of 500 ng, using jetPEI polyplus reagent according to manufacturer's instructions. 48 hr later, cells were trypsinised and transferred into 10 cm dish, and ouabaine (sigma) was added to a final concentration of 0.5 µM, for one week. Clones were then recovered and screened by PCR for the presence of Flag-tag. Positive clones were verified by sequencing. Primers for each guide are detailed in *Supplementary file 13*.

## Antibodies and immunoblotting
Total Cell extracts were prepared and analysed by standard Western blotting protocol, using antibodies against total H2AZ (ab4174, Abcam), GAPDH (clone 6C5, MAB374, Millipore), PHF14 (SAB3500960, Sigma and proteintech 24787–1-AP), SIRT1 (Clone E104, ab32441, Abcam), Flag (Clone M2, F1804, Sigma), Pol I (Clone RPA194, sc48385, Santa Cruz), H3 (ab1791, Abcam), pan Acetyl-H3 (6599, Upstate), Acetyl-H3K9 (06942, Upstate), Flag-HRP (Sigma, A8592, lot #GR08726011-2013), Brd8 (Bethyl A300-219A), DMAP1 (Aff. BioReag., PA1-886), ACTR6(ARP6) (Abcam, ab208830), p400 (Abcam, ab5201), SRCAP (gift from J. Chrivia), BAF53a (Abcam, ab3882, lot #9118237), EPC1 (Abcam, ab5514, lot#98723).

## Tandem-affinity purification of endogenous H2AZ.1 and H2AZ.2
Purification of endogenously tagged H2AZ.1 and H2AZ.2 was performed basically as described (*Doyon and Côté, 2016*). Typically, soluble nuclear extracts (*Abmayr et al., 2006*) were prepared from 3E9 cells (3 L cultures at 0.6–1.0 million cells per ml), adjusted to 0.1% Tween-20, and centrifuged at 100,000 g for 45 min. Extracts were precleared with 300 ul Sepharose CL-6B (Sigma), then 250 ul anti-FLAG M2 affinity resin (Sigma) was added for 2 hr at 4°C. The beads were then washed in Poly-Prep columns (Bio-Rad) with 40 column volumes (CV) of buffer #1 (20 mM HEPES-KOH).

[pH 7.9], 10% glycerol, 300 mM KCl, 0.1% Tween 20, 1 mM DTT, 1 mM PMSF, 2 µg/mL Leupeptin, 5 µg Aprotinin, 2 µg/mL Pepstatin, 10 mM Na-butyrate, 10 mM β-glycerophosphate, 100 µM Na-orthovanadate, 5 mM N-Ethylmaleimide, 2 mM Ortho-Phenanthroline) followed by 40 CV of buffer #2 (20 mM HEPES-KOH [pH 7.9], 10% glycerol, 150 mM KCl, 0.1% Tween 20, 1mMDTT, 1 mM PMSF, 2 µg/mL Leupeptin, 5 µg Aprotinin, 2 µg/mL Pepstatin, 10 mM Na-butyrate, 10 mM β-glycerophosphate, 100 µM Na-orthovanadate, 5 mM N-Ethylmaleimide, 2 mM Ortho-Phenanthroline). Complexes were eluted in two fractions with 2.5 CV of buffer #2 supplemented with 200 ug/ml 3xFLAG peptide (Sigma) for 1 hr at 4°C. Next, fractions were mixed with 125 ul Strep-Tactin Sepharose (IBA) affinity matrix for 2 hr at 4°C, and the beads were washed with 20 CV of buffer #2. Complexes were eluted in two fractions with 2 CV of buffer #2 supplemented with 4 mM D-biotin, flash frozen in liquid nitrogen, and stored at −80°C. Typically, 15 ul of the first elution (3% of total) was loaded on NuPAGE 4–12% Bis-Tris gels (Invitrogen) and analyzed by silver staining.

## Mass-spectrometry analysis
The analyses were performed at the proteomic platform of the Quebec Genomics Center.

The peptides were directly loaded at 300 nL/min onto a New Objective PicoFrit column (15 cm ×0.075 mm I.D; Scientific Instrument Services, Ringoes, NJ) packed with Jupiter 5 µm $C_{18}$ (Phenomenex, Torrance, CA) stationary phase. The peptides were eluted from the column by a gradient generated by an Agilent 1200 HPLC system (Agilent, Santa Clara, CA) equipped with a nano electrospray ion source coupled to a 5600+ Triple TOF mass spectrometer (Sciex, Concord, ON). A 65 min linear gradient of a 5–35% mixture of acetonitrile, 0.1% formic acid injected at 300 nL/min was used to elute peptides. Data dependent acquisition mode was used in Analyst version 1.7 (Sciex) to acquire mass spectra. Full scan mass spectrum (400 to 1250 m/z) were acquired and followed by collision-induced dissociation of the twenty most intense ions. A period of 20 s and a tolerance of 100 ppm were set for dynamic exclusion.

Protein Pilot version 5.0 (Sciex) was used to generate MS/MS peak lists. Mascot (Matrix Science, London, UK; version 2.4.0) was used to analyze MGF sample files. Mascot was set up to search the UniprotKB *Homo sapiens* database (release 11/2014, 162831 sequences) assuming the digestion enzyme trypsin. Mascot was searched with a fragment ion mass tolerance and a parent ion tolerance of 0.1 Da. Oxidation of methionine and deamidation of asparagine and glutamine were specified as variable modifications and carbamidomethylation as fixed modification. Two missed cleavages were allowed. Scaffold (version 4.0.1), Proteome Software Inc, Portland, OR) was used to validate MS/MS based peptide and protein identifications. Proteins/peptides FDR rate was set to 1% or less based on decoy database searching. The Protein Prophet algorithm assigned protein probabilities. Proteins that contained similar peptides and could not be differentiated based on MS/MS analysis alone were grouped to satisfy the principles of parsimony.

Data were further analyzed using the CRAPome online tool (www.crapome.org; *Mellacheruvu et al., 2013*) with SAINTexpress default parameters, and visualized using ProHits-viz (https://prohits-viz.lunenfeld.ca/; *Knight et al., 2017*). Two replicates were used for each bait and normalized on Histone H2A.Z-H2B chaperones (histones themselves have too low spectral counts).

## Cell fractionation
### Cell fractionation using NP40
Cell pellets (5 million cells) were resuspended in 250–300 µl of lysis buffer (10 mM Tris pH 8.0, 10 mM NaCl, 2 mM MgCl2) and incubated at 4°C for 5 min. 10 µl were kept to prepare whole cell extracts. 50 µl/ml of NP40 were then added and samples were incubated for additional 10 min at 4°C and then centrifuged at 3000 g for 5 min. The supernatant was collected and represented the cytoplasmic fraction. The remaining pellet was resuspended in 35 µl of nuclear buffer (20 mM Hepes pH 7.9, 150 mM NaCl, 1.5 mM MgCl2, 0.2 mM EDTA, 10% Glycerol) and incubated 30 min at 4°C. Samples were centrifuged at full speed for 5 min. The supernatant was removed and a second extraction was performed on the remaining pellet with a nuclear buffer high in salt (20 mM Hepes pH 7.9, 420 mM NaCl, 1.5 mM MgCl2, 0.2 mM EDTA, 10% Glycerol). After 30 min of incubation at 4°C, samples were centrifuged at full speed. The supernatant was collected and represented the soluble nuclear fraction. The pellet corresponding to the chromatin was resuspended in boiling buffer (1% SDS, 1% Triton, 10 mM Tris PH 7.4, 0.5 M NaCl) and sonicated 5 times at 25% amplitude for 10 s. Whole cell extracts were also resuspended in boiling buffer and sonicated. All the buffers were supplemented with EDTA-free protease inhibitor cocktail (Roche).

### Cell fractionation under mild conditions
Cells were trypsinated, counted, and resuspended at a concentration of $4 \times 10^7$ cells/ml in buffer A (10 mM HEPES, 10 mM KCl, 1.5 mM MgCl2, 0.34 M Sucrose, 10% glycerol, 0.1% Triton X-100 and 1 mM DTT) and incubated at 4°C for 8 min. Small volume was kept to prepare whole cell extracts. The samples were then centrifugated at 1300 g for 5 min. The supernatant was collected and represented the cytoplasmic fraction. The nuclei pellet was washed once in buffer A and lysed for 30 min on ice in buffer B (3 mM EDTA, 0.2 mM EGTA and1 mM DTT). Samples were centrifuged at full speed for 5 min. The supernatant was collected and represented the soluble nuclear fraction. The pellet corresponding to the insoluble chromatin was washed once in buffer B. The chromatin was then extracted with buffer A and sonicated 3 times at 25% amplitude for 10 s. Whole cell extracts were sonicated with the same protocole. All frations were clarified by full speed centrifugation. The buffers were supplemented with EDTA-free protease inhibitor cocktail (Roche).

## RNA extraction and reverse transcription

Total RNA was prepared using the MasterPure RNA Purification Kit (Epicentre) supplemented with Baseline-ZERO DNase, according to the manufacturer's instructions. For random-primed RT-qPCR, 200 ng of RNA were used for each reverse transcription reaction. The reverse-transcription was performed using random primers and superscript III reverse transcriptase (Invitrogen) at 50°C according to manufacturer's protocol. qPCR analysis was performed on CFX96 devices (BioRad) using the SYBR Premix Ex Taq II (Takara), according to the manufacturer's instructions. All samples was analysed in duplicates. All data was normalized relative to GAPDH mRNA levels. The list of primers can be found in *Supplementary file 13*.

## ChIP experiments

15 million cells transfected with siRNA were crosslinked for 15 min using 1% formaldehyde directly in the culture medium. 0.125 M of Glycine were then added for 5 min. After two washes with PBS, cells were scraped and frozen at −80°C. Cells were lysed with 3 ml of a lysis buffer (5 mM Pipes PH 8, 85 mM KCl, 0.5% NP40) and homogenized 40 times with a dounce (20 times, pause 2 min, 20 times). After centrifugation, nuclear pellets were resuspended in 1.5 ml of nuclear lysis buffer (50 mM Tris PH 8.1, 10 mM EDTA, 0.2% SDS), and sonicated 10 times for 10 s (power setting 5% and 50% duty cycle, Branson Sonifier 250), to obtain DNA fragments of about 500 bp. DNA concentration was determined using a Nanodrop and samples were adjusted to the same concentration of chromatin. Samples were diluted at least one time in dilution buffer (0.01% SDS, 1.1% Triton, 1.2 mM EDTA, 16.7 mM Tris pH 8.1, 167 mM NaCl) and precleared for 2 hr with 250 μl of previously blocked protein-A and protein-G beads (Sigma P-7786 and P-3296 respectively). Blocking was achieved by incubating the beads with 0.5 mg/ml of Ultrapure BSA for 3 hr at 4°C. ChIP reaction was performed in 1 ml final volume. 100 μl of chromatin were kept for inputs. 50 μg of pre-cleared samples per ChIP supplemented with 10 ng of *Drosophila melanogaster* chromatin (spike in chromatin, Active motif), and 1 ug of an antibody recognizing H2Av, a *Drosophila* specific histone variant, (spike in antibody, active motif), were incubated overnight with 4 μg of antibody at 4°C. A mock sample without antibody was processed similarly. Then, 20 μl of blocked A/G beads were added for 2 hr at 4°C to recover immune complexes. Beads were washed once in dialysis buffer (2 mM EDTA, 50 mM Tris pH 8, 0.2% Sarkosyl), four times in wash buffer (100 mM Tris pH 8.8, 500 mM LiCl, 1% NP40, 1% NaDoc) and twice in TE buffer (10 mM Tris pH 8, 1 mM EDTA). The bead/chromatin complexes were resuspended in 200 μl of TE buffer and incubated 30 min at 37°C with 10 μg of RNase A (Abcam), as well as input DNA. Formaldehyde crosslink was reversed in the presence of 0.2% SDS at 70°C overnight with shaking. After 2 hr of proteinase K (0.2 mg/ml) treatment at 45°C, immunoprecipitated and input DNA were purified on columns using Illustra GFX kit (GE Healthcare). All buffers for ChIP experiment were supplemented with EDTA-free protease inhibitor cocktail (Roche) and filtered 0.2 μM. Results were analysed by qPCR. The list of primers used can be found in *Supplementary file 13*.

## Immunoprecipitation

8 million cells were lysed with 600 μl the lysis buffer (10 mM Tris PH 8, 1% NP40, 420 mM NaCl, 2 mM EDTA). Samples were incubated 15 min on ice and vortexed each 5 min. After 15 min of centrifugation at full speed, supernatant was recovered and diluted 3 times with dilution buffer (20 mM Tris PH 8, 2 mM EDTA, 20 μl DNase I epicenter, 25 mM CaCl2). NP40 final concentration was adjusted to 0.5%, and lysates were quantified. 1 mg was used per immunoprecipitation reaction. Preclearing was done by incubating lysates 2 hr at 4°C with 15 μl of A/G beads (Sigma). 4 μg of Flag antibody were then added ON at 4°C. 15 μl of A/G beads were added for 2 hr at 4°C, and immunocomplexes were washed 4 times with wash buffer (20 mM Tris PH 8, 0.5% NP40, 140 mM NaCl, 2 mM EDTA). Results were analysed by Western blot.

## RNA-Seq

In WI38 cells, to identify the interplay between H2A.Z. isoforms, two samples of siCtrl, siH2A.Z.2, and siH2A.Z.1 + siH2A.Z.2 and one sample of siH2A.Z. one were transfected in parallel. The other siH2A.Z.1 sample came from *Muniz et al. (2017)*. To identify genes regulated by PHF14 and SIRT1, two samples of sictrl, siPHF14 and siSIRT1 were transfected in parallel.

In U2OS cells, two samples of siCtrl, two sample of si Ctrl#, siH2A.Z.1, siH2A.Z.2 and siH2A.Z.1 +siH2A.Z.2 were transfected in parallel.

We used strand-specific RNA-Seq method, relying on UTP incorporation in the second cDNA strand. For each sample, 5–10 µg of total RNA, extracted as described above in (RNA extraction and Reverse transcription), was submitted to EMBL-GeneCore, Heidelberg, Germany. Paired-end sequencing was performed by Illumina's NextSeq 500 technology. Two replicates of each sample were sequenced.

## ChIP-seq

For the ChIP-seq in U2OS cells expressing Flag-H2AZ.1 and Flag-H2AZ.2, 100 µg of chromatin supplemented with 10 ng of spike in chromatin (active motif) were used per ChIP experiment. For each reaction, 4 ug of Flag M2 antibody (sigma) and 1 ug of spike in antibody (active motif), were used. About 10 ng of immunoprecipated DNA (quantified with quantiFluor dsDNA system, Promega) were obtained at each time, and samples were submitted to EMBL-GeneCore Heidelberg for sequencing, that was performed by Illumina's NextSeq 500 technology.

ChIP-seq in K562 cells expressing Flag-H2A.Z.1 and Flag-H2A.Z.2 were performed and analysed as previously described (*Jacquet et al., 2016*; *Lalonde et al., 2013*).

## RNA-Seq processing

RNA-Seq samples were sequenced using Illumina NextSeq 500 sequencer, paired-end, 80 bp reads, at EMBL Genomics core facilities (Heidelberg, Germany). The quality of each raw sequencing file (fastq) was verified with FastQC (*Andrews, 2010*). Files were aligned to the reference human genome (hg38) in paired-end mode with STAR Version 2.5.2b (*Agudelo et al., 2017*) and processed (sorting and indexing) with samtools (*Li et al., 2009*). Raw reads were counted, per gene_id, using HT-seq Version 0.6.1 (*Anders et al., 2015*) on the NCBI refseq annotation gtf file from UCSC in a strand specific mode with default parameters.

## RNA-Seq analysis and figures

Several differential analysis (siH2A.Z.1 vs siCtrl, siH2A.Z.2 vs siCtrl, siH2A.Z1 and siH2A.Z.2 vs siCtrl, siSIRT1 vs siCtrl, siPHF14 vs siCtrl) were done with DESeq2 Bioconductor R package, Version 1.22.1 (*Love et al., 2014*) with default parameters. In U2OS cells, four control samples obtained using two different siRNAs were used. Genes of interest were selected when |log2FoldChange| higher $\log_2(1.25)$ and adjusted p-value lower than 0.05. RPKM shown in *Supplementary files 1–12* were calculated for each gene in the different datasets by taking the raw counts from HT-seq multiplied by 1E+09/(total number of aligned reads * sum of exons' sizes of the gene).

Lists of gene of interest were crossed to identify common genes, and results were represented using Venn Diagrams with R and VennDiagram package. For each crossing, the Chi-square test (chisq.test() function in R) was applied to the associated contingency table. The test evaluate if the 2 lists of genes are independent that is whether there is a significant association between the categories of the two variables. Chi-square p-values were then corrected for multiple testing (one test per crossing) with the Benjamini–Hochberg method.

Box-plots representing the RPKM value were generated with R-base based on the mean of replicates from Tables S1-8. The center line represents the median, box ends represent respectively the first and third quartiles, whiskers represent the minimum and maximum values without outliers. Outliers were defined as below 1stQuartile − 1.5 × InterquartileRange and above 3rdQuartile + 1,5 × InterquartileRange. Nonparametric Mann–Whitney–Wilcoxon test (wilcoxon.test() function in R) was applied to test distribution differences between two populations.

## ChIP-Seq processing and analysis

ChIP-Seq samples were sequenced using Illumina NextSeq 500 sequencer, single-end, 80 bp reads, at EMBL Genomics core facilities (Heidelberg, Germany).

The quality of each raw sequencing file was verified with FastQC. Files were aligned to the reference human genome (hg38) in single-end mode with (*Li and Durbin, 2009*) and processed (sorting, PCR duplicates removing and indexing) with samtools. The coverage was computed with the GenomicAlignments Bioconductor R package (*Lawrence et al., 2013*).

ChiP-Seq mean coverage per base was computed for each annotated gene, in a window of +/- 2 kb around Transcription Start Site (TSS). For each gene, the log2 mean value in these windows was computed and plotted using R-base, in a dot-plot representing H2AZ1 versus H2AZ2. Lastly, the log2 ratio of the mean value in these windows for H2AZ1 divided by the mean value in these windows for H2AZ2 was computed for 5 list of genes selected through the RNA-Seq differential analysis: genes up-regulated from siH2AZ1 versus siCtle, genes down-regulated from siH2AZ1 versus siCtle, genes up-regulated from siH2AZ2 versus siCtle, and un-regulated genes in both analysis. Results of these calculations were shown in a boxplot.

## Acknowledgements

The authors wish to thank Samer Hussein for bioinfo/metagene analyses of K562 ChIP-seq, Lisa Muniz and all members of DT's and JC's teams for helpful discussions, and Jean-Philippe Lambert for help with mass spectrometry data management.

## Additional information

### Funding

| Funder | Grant reference number | Author |
|---|---|---|
| Canadian Institutes of Health Research | FDN-143314 | Jacques Cote |
| Fondation ARC pour la Recherche sur le Cancer | Programme ARC | Didier Trouche |
| Ligue Contre le Cancer | Equipe labellisée | Didier Trouche |
| Fondation pour la Recherche Médicale | Bourse de post-doc | Assala Lamaa |

The funders had no role in study design, data collection and interpretation, or the decision to submit the work for publication.

### Author contributions

Assala Lamaa, Conceptualization, Methodology, Investigation, visualization; Jonathan Humbert, Conceptualization, Methodology, Investigation; Marion Aguirrebengoa, Investigation, Visualization, Methodology; Xue Cheng, Methodology, Investigation; Estelle Nicolas, Resources, Visualization, Methodology; Jacques Côté, Conceptualization, Supervision, Funding acquisition; Didier Trouche, Conceptualization, Supervision, Funding acquisition, Project administration

### Author ORCIDs

Assala Lamaa ⬤ https://orcid.org/0000-0001-7706-1694
Jonathan Humbert ⬤ https://orcid.org/0000-0002-1090-6207
Jacques Côté ⬤ https://orcid.org/0000-0001-6751-555X
Didier Trouche ⬤ https://orcid.org/0000-0003-1398-6481

### Decision letter and Author response

Decision letter https://doi.org/10.7554/eLife.53375.sa1
Author response https://doi.org/10.7554/eLife.53375.sa2

## Additional files

### Supplementary files

- Supplementary file 1. Genes upregulated upon H2A.Z.1 depletion in WI38 cells.
- Supplementary file 2. Genes upregulated upon H2A.Z.2 depletion in WI38 Cells.
- Supplementary file 3. Genes down-regulated upon H2A.Z.1 depletion in WI38 cells.

- Supplementary file 4. Genes down-regulated upon H2A.Z.2 depletion in WI38 cells.
- Supplementary file 5. Genes regulated upon the combined depletion of H2A.Z.1 and H2A.Z.2 in WI38 cells.
- Supplementary file 6. Genes upregulated upon H2A.Z.1 depletion in U2OS cells.
- Supplementary file 7. Genes upregulated upon H2A.Z.2 depletion in U2OS Cells.
- Supplementary file 8. Genes down-regulated upon H2A.Z.1 depletion in U2OS cells.
- Supplementary file 9. Genes down-regulated upon H2A.Z.2 depletion in U2OS cells.
- Supplementary file 10. Genes regulated upon the combined depletion of H2A.Z.1 and H2A.Z.2 in U2OS cells.
- Supplementary file 11. Genes regulated upon PHF14 depletion in WI38 cells.
- Supplementary file 12. Genes regulated upon SIRT1 depletion in WI38 cells.
- Supplementary file 13. List of siRNA and primers.
- Transparent reporting form

## Data availability

Deep Sequencing Data are available at GEO (accession number: # GSE131579). MS and scaffold files generated in this study were deposited at MassIVE (http://massive.ucsd.edu) and assigned the MassIVE accession numbers MSV000084836. Source data files have been added for all histograms.

The following datasets were generated:

| Author(s) | Year | Dataset title | Dataset URL | Database and Identifier |
|---|---|---|---|---|
| Lamaa A, Humbert J, Aguirrebengoa M, Xue C, Nicolas E, Coté J, Trouche D | 2020 | Integrated analysis of H2A.Z isoforms functions reveals a complex interplay in gene regulation. | https://www.ncbi.nlm.nih.gov/geo/query/acc.cgi?acc=GSE131579 | NCBI Gene Expression Omnibus, GSE131579 |
| Lamaa A, Humbert J, Aguirrebengoa M, Xue C, Nicolas E, Coté J, Trouche D | 2020 | Integrated analysis of H2A.Z isoforms functions reveals a complex interplay in gene regulation. | ftp://MSV000084836@massive.ucsd.edu | MassIVE, MSV000084836. |

The following previously published dataset was used:

| Author(s) | Year | Dataset title | Dataset URL | Database and Identifier |
|---|---|---|---|---|
| Greenberg RS, Long HK, Swigut T, Wysocka J | 2019 | Single Amino Acid Change Underlies Distinct Roles of H2A.Z Subtypes in Human Syndrome. | https://www.ncbi.nlm.nih.gov/geo/query/acc.cgi?acc=GSE134532 | NCBI Gene Expression Omnibus, GSE134532 |

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
