## [Decision Letter]

**Acceptance summary:**

This paper begins to resolve the previously enigmatic roles in gene regulation of two closely related isoforms of the variant histone H2A.Z. By engineering cells to allow the targeted depletion of one or both of these isoforms, the authors show, surprisingly, that these two nearly identical proteins work antagonistically, and make progress towards understanding the molecular basis of this interaction. The antagonism of two forms of the same histone variant/reader combination represents a novel mechanism for modulating gene activity, and should be of broad interest.

**Decision letter after peer review:**

Thank you for submitting your article "Integrated analysis of H2A.Z isoforms function reveals a complex interplay in gene regulation" for consideration by *eLife*. Your article has been reviewed by two peer reviewers, and the evaluation has been overseen by a Michael Eisen as the Senior and Reviewing Editor. The reviewers have opted to remain anonymous.

The reviewers have discussed the reviews with one another and the Reviewing Editor has drafted this decision to help you prepare a revised submission.

The manuscript deals with the functional analysis of the two histone variant H2A.Z isoforms. These isoforms are highly similar with only three amino acid differences, but have been recently shown to play distinct roles in tumor development, gene regulation and DNA repair. The authors have used siRNAs against edited epitope-tagged versions of the isoforms to disentangle their individual roles in gene regulation. H2A.Z occupies nucleosomes that flank active promoters and enhancers and has long been enigmatic in its effects on gene regulation, showing up-regulation at some genes and down-regulation at others. Here this finding is confirmed by isoform depletions in two different human cell lines, but the unexpected result is that depletion of both isoforms results in effects opposite to expectation, namely depletion of both isoforms restores a more wild-type-like situation.

Overall, the reviewers judged that the authors made important observations that are of interest for the histone variant and chromatin research field. There was some discussion of whether additional replicates and therefore more robust statistical analysis is required. Recognizing that doing such experiments might take more than 2 months and would therefore be inconsistent with *eLife* policy not to request such experiments, we are leaving the inclusion of additional data to the authors' discretion.

The reviewers made specific comments and questions listed below that we request be addressed in the revision.

Essential revisions:

In Figures 2 and 6 Venn diagrams are shown to indicate the number of genes showing effects caused by knockdown of either isoform and the overlap indicates the number of genes affected by both, with p-values based on Chi-square tests. However, astronomical p-values are the norm for such comparisons involving changes in thousands of genes, and the p-values are potentially sensitive to minor technical biases. Also p-values compare observations to random expectation, which is not the same as asking how different the results are from what no treatment would show. However, each experiment included a control siRNA, and so the degree to which technical variation might have inflated these p-values can be estimated by comparing each to its siRNA control as was done in the histograms and box-plots.

It is not always clear how many replicates are being analyzed, or what kind of statistics are being used to report on the data (c.f. Figure 1C). In some places it appears that error bars are being used for two samples.

Figure 1B: What is the band observed at around 20 kDa? Could this be modified, possibly ubiquitinated or sumoylated, endogenous protein? In this case, is such a shift also observed with the Flag-tagged histones? A larger view of the blot would be helpful, as well as blots with a Flag antibody.

Why do the authors use K562 cells instead of the in the study well-characterized U2OS cells for protein-pull-down assays? Do K562 cells show a similar effect in deregulated genes upon H2A.Z.1 or H2A.Z.2 depletion?

What is the experimental difference compared to previously performed mass-spectrometry-based identifications of H2A.Z interactors? The authors should indicate in the Results section whether they used nuclear extracts, chromatin-fractions, mononucleosomes, whole cell extracts or others. Why were the authors not able to identify direct and strong H2A.Z interacting proteins such as e.g. BRD2 (Draker et al., 2012), PWWP2A (Pünzeler et al., 2017), but were able to identify the p400 chaperone complex that interacts mostly with free nuclear H2A.Z proteins? Did the authors use chromatin-free nuclear extracts?

Figure 5—figure supplement 1C: The figure legend is not sufficient to describe the experimental data in a way that it is understandable. What does NT mean? Not transfected, no tag? It would be good to show where the primers bind to explain the size differences between H2AFZ and H2AFV PCR products. Ideally, the authors would include western blot analysis with H2A.Z antibody for these cells as done for U2OS cells in Figure 1.

Figure 5—figure supplement 1E: The legend describes 5 classes, shown are 4 groups labelled with G1-G4. Are these groups labelled according to gene expression levels: G1 is lowest and G4 highest? More information is required to understand the figure.

Figure 6: the authors should indicate how many times these experiments were repeated. In A) H2A.Z western blots should be shown to document knock-down efficiency. Also, H2A.Z.1 seems to lead to an increase in Flag-PHF14 while H2A.Z.2 depletion to a decrease. The authors should quantify this result and, if reproduced, check whether PHF14 RNA transcripts are also affected. Hence, more experiments are required to strengthen the statement: "Thus H2A.Z.1 favours the interaction of PHF14 with chromatin", especially since there is no difference observed in PHF14 levels in the chromatin fraction upon H2A.Z.1 depletion. Similar for SIRT1 (Figure 6B).

Figure 6C and Figure 1D: when comparing the effects of PHF14 or SIRT1 depletion on the expression of those six gene with the effects observed with H2A.Z.1 or H2A.Z.2 depletion, I do not agree with the conclusions made by the authors in the subsection “PHF14 and SIRT1 are major H2A.Z1 and H2A.Z.2 effectors”. The authors should use the same axis values to better compare the results. It appears that only the PLAT and AKAP12 genes shows a similar trend, all other genes look different. ZDHHC20 shows only minimal increase when PHF14 is depleted (4-fold in siZ1 conditions), RRM2 is not much lower in siSIRT1 conditions, but half the amount in siZ2, COLEC12 and ADAMTS1 do not show any change, but 3/4-fold increases in siZ2 treatments. Also, while there is some overlap of deregulated genes in Figure 6D, E, the statement "Thus, taken together, these data indicate that PHF14 and SIRT1 are major mediators of H2A.Z.1- and H2A.Z.2-specific gene regulation…" is an over-interpretation of the data. The conclusion should be toned down or other data should be included that strengthen this statement.

Figure 7: Additional replication would be ideal for 7B and additional targets, including ones not dependent on isoform, for 7C.

The authors state: "The conclusion from these experiments is that PHF14 and SIRT1 could have antagonistic effect on local protein acetylation balance, providing a molecular mechanism by which PHF14 and the SIRT1 deacetylase can mediate H2A.Z.1 and H2A.Z.2 antagonism". This is a too strong conclusion, as no experiments have been performed to show that H3K9ac changes upon PHF14 depletion are due to SIRT1 activity at those sites. Many other proteins can contribute to the observed effects. In order to make such a statement, the authors need to perform PHF14 and SIRT1 ChIP-seq experiments to show their localization at H2A.Z sites and how their localization changes when H2A.Z.1, H2A.Z.2, PHF14 or SIRT1 are depleted. Hence, I would also remove the model in Figure 7D.

Discussion: This is not the first study showing that H2A.Z.1 and H2A.Z.2 bind differentially to proteins. Draker et al. already showed a preference of BRD2 binding to H2A.Z.1, an observation also confirmed by Pünzeler et al. Further, in Pünzeler et al., PHF14, HMG20A, TCF20 and RAI1 were seen to be enriched on H2A.Z.2 nucleosomes when compared to H2A.Z.1 nucleosomes, as was also seen for other interacting proteins. The authors should rephrase their conclusions.

---

## [Author Response]

Essential revisions:In Figures 2 and 6 Venn diagrams are shown to indicate the number of genes showing effects caused by knockdown of either isoform and the overlap indicates the number of genes affected by both, with p-values based on Chi-square tests. However, astronomical p-values are the norm for such comparisons involving changes in thousands of genes, and the p-values are potentially sensitive to minor technical biases. Also p-values compare observations to random expectation, which is not the same as asking how different the results are from what no treatment would show. However, each experiment included a control siRNA, and so the degree to which technical variation might have inflated these p-values can be estimated by comparing each to its siRNA control as was done in the histograms and box-plots.

First, we want to point out that, in RNA Seq experiments following each protein depletion, all samples were performed together, and two control samples were always included. Differentially expressed genes were then identified compared to these two control samples (i.e., for example, PHF14 and SIRT1 RNA-Seq were analyzed compared to two controls transfected in parallel, which are different from the two controls used for H2A.Z isoforms depletion. We modified the Materials and methods section to clarify this point.

We next tested whether the regulation of genes after H2A.Z.1 depletion is independent, or not, with the regulation of genes after H2A.Z.2 depletion. We also tested whether the regulation of genes after H2AZ.1 or H2AZ.2 depletion is independent, or not, with the regulation of genes after PHF14 or SIRT1 depletion, respectively. The Chi-square testprovides a method for testing the association between 2 variables in a two-way table. The null hypothesis H0 assumes that there is no association between the 2 variables (In other words, one variable does not vary according to the other variable), while the alternative hypothesis H1 claims that some association does exist. In all the dependencies we tested (except one shown in Figure 2—figure supplement 3B), the tests indicated that we can reject the H0 hypothesis.

However, we do agree with the reviewer that the Chi-square test gives astronomical p values when analyzing large populations. As the reviewer says, it compares to random expectations. In order to clarify how different the result is from random, we now indicate in all Venn diagrams of the revised manuscript the expected number of common genes in the case of random intersection, providing the quantitative data requested by the reviewer.

It is not always clear how many replicates are being analyzed, or what kind of statistics are being used to report on the data (c.f. Figure 1C). In some places it appears that error bars are being used for two samples.

We changed the manuscript to carefully state this information in figure legends. Note that in the original manuscript (as well as in data added for revision), all histograms presenting qPCR data were calculated on three entirely independent experiments. The only exception were histograms comparing RT-qPCR with RNA Seq data (now Figure 2—figure supplement 1), given that in this figure, we analyzed by RT-qPCR the two experiments used for RNA-Seq. We now show the RT-qPCR results from the two samples side-by-side.

The only other experiment where error bars were calculated using two replicates was the quantification of the western blot shown in Figure 1C. We now present a representative experiment out of two.

Figure 1B: What is the band observed at around 20 kDa? Could this be modified, possibly ubiquitinated or sumoylated, endogenous protein? In this case, is such a shift also observed with the Flag-tagged histones? A larger view of the blot would be helpful, as well as blots with a Flag antibody.

As requested by the reviewer, we performed a Flag western blot both in U2OS and K562 cells. These data indicate that in these two cell lines this band is present, confirming that it is indeed produced from endogenous H2A.Z, and that it is observed with the two isoforms. It most likely corresponds to ubiquitinated H2A.Z, in a ratio that has been reported before in the literature. This Flag western blot in U2OS cells is shown in the revised manuscript in Figure 1—figure supplement 1 and the Flag western blot in K562 cells is shown in Figure 5—figure supplement 1. The text of the manuscript has been changed accordingly.

Why do the authors use K562 cells instead of the in the study well-characterized U2OS cells for protein-pull-down assays? Do K562 cells show a similar effect in deregulated genes upon H2A.Z.1 or H2A.Z.2 depletion?

We use K562 cells because they are nonadherent cells which are easier to produce in large amounts for endogenous nuclear proteins purifications and mass spectrometry analysis of interacting proteins. This is now stated in the manuscript.

In the time allowed during revision, we confirmed the preferred interactions between H2A.Z isoforms with SIRT1 or PHF14 in U2OS cells by western blotting after affinity purification. These data have been included in the revised manuscript in Figure 5—figure supplement 2.

What is the experimental difference compared to previously performed mass-spectrometry-based identifications of H2A.Z interactors? The authors should indicate in the Results section whether they used nuclear extracts, chromatin-fractions, mononucleosomes, whole cell extracts or others. Why were the authors not able to identify direct and strong H2A.Z interacting proteins such as e.g. BRD2 (Draker et al., 2012), PWWP2A (Pünzeler et al., 2017), but were able to identify the p400 chaperone complex that interacts mostly with free nuclear H2A.Z proteins? Did the authors use chromatin-free nuclear extracts?

The reviewer is right in thinking that we were using chromatin-free nuclear extracts (Dignam classical 300mM salt soluble extraction from nuclei). This was actually indicated in the Materials and methods section, but we agree with the reviewer that it is important. Indeed, it explains why we detect the p400, SRCAP and chaperone complexes but not BRD2 and PWWP2A proteins which were identified in over-expressed H2A.Z nucleosome pull-downs from digested chromatin fractions. Accordingly our mass spec data indicates a majority of H2A.Z-H2B dimers compared to octamers. In the revised manuscript, we modified the Results section to indicate the different purification approach compared to what has been reported in other studies. At the same time our different approach provides a new distinct interactome dataset, in a completely physiological setting since we purified endogenous H2A.Z isoforms.

Figure 5—figure supplement 1C: The figure legend is not sufficient to describe the experimental data in a way that it is understandable. What does NT mean? Not transfected, no tag? It would be good to show where the primers bind to explain the size differences between H2AFZ and H2AFV PCR products. Ideally, the authors would include western blot analysis with H2A.Z antibody for these cells as done for U2OS cells in Figure 1.

We changed the figure to specify what NT is and included western blot data as requested by the reviewer (Figure 5—figure supplement 1 of revised manuscript). Primers were already used in Figure 1—figure supplement 1, which is now referred to in the legend.

Figure 5—figure supplement 1E: The legend describes 5 classes, shown are 4 groups labelled with G1-G4. Are these groups labelled according to gene expression levels: G1 is lowest and G4 highest? More information is required to understand the figure.

We apologize for this mistake. We corrected it in Figure 5—figure supplement 1 from revised manuscript.

Figure 6: the authors should indicate how many times these experiments were repeated. In A) H2A.Z western blots should be shown to document knock-down efficiency.

This information has now been included in the revised manuscript. H2A.Z western blot is shown in Figure 6—figure supplement 1.

Also, H2A.Z.1 seems to lead to an increase in Flag-PHF14 while H2A.Z.2 depletion to a decrease. The authors should quantify this result and, if reproduced, check whether PHF14 RNA transcripts are also affected. Hence, more experiments are required to strengthen the statement: "Thus H2A.Z.1 favours the interaction of PHF14 with chromatin", especially since there is no difference observed in PHF14 levels in the chromatin fraction upon H2A.Z.1 depletion. Similar for SIRT1 (Figure 6B).

No effect could be seen by RT-qPCR. This is now stated in the manuscript. The results are reproducible and we now show two independent experiments (Figure 6 and Figure 6—figure supplement 2). In addition, we now show in Figure 6—figure supplement 2 a quantification showing that the repartition of PHF14 between chromatin and soluble nuclear fraction changes upon H2A.Z1 depletion, therefore suggesting that the presence of H2A.Z.1 favours chromatin localisation of PHF14. We analysed and present similarly SIRT1 data. However, we agree with the reviewer that more experiments would be needed to strengthen this point, and we toned down our conclusions when describing these data.

Figure 6C and Figure 1D: when comparing the effects of PHF14 or SIRT1 depletion on the expression of those six gene with the effects observed with H2A.Z.1 or H2A.Z.2 depletion, I do not agree with the conclusions made by the authors in the subsection “PHF14 and SIRT1 are major H2A.Z1 and H2A.Z.2 effectors”. The authors should use the same axis values to better compare the results. It appears that only the PLAT and AKAP12 genes shows a similar trend, all other genes look different. ZDHHC20 shows only minimal increase when PHF14 is depleted (4-fold in siZ1 conditions), RRM2 is not much lower in siSIRT1 conditions, but half the amount in siZ2, COLEC12 and ADAMTS1 do not show any change, but 3/4-fold increases in siZ2 treatments. Also, while there is some overlap of deregulated genes in Figure 6D, E, the statement "Thus, taken together, these data indicate that PHF14 and SIRT1 are major mediators of H2A.Z.1- and H2A.Z.2-specific gene regulation…" is an over-interpretation of the data. The conclusion should be toned down or other data should be included that strengthen this statement.

We agree with this reviewer that phenocopying was an over-interpretation. As suggested by this reviewer, we now use the same axes in Figure 6C and 1D (now Figure 2—figure supplement 1) and have now toned down our conclusion. Indeed, as indicated above, H2A.Z1 probably regulates genes independently of PHF14, and vice-versa.

Figure 7: Additional replication would be ideal for 7B and additional targets, including ones not dependent on isoform, for 7C.

We performed the requested experiments and included them in the manuscript (Figure 7 and Figure 7—figure supplement 1).

The authors state: "The conclusion from these experiments is that PHF14 and SIRT1 could have antagonistic effect on local protein acetylation balance, providing a molecular mechanism by which PHF14 and the SIRT1 deacetylase can mediate H2A.Z.1 and H2A.Z.2 antagonism". This is a too strong conclusion, as no experiments have been performed to show that H3K9ac changes upon PHF14 depletion are due to SIRT1 activity at those sites. Many other proteins can contribute to the observed effects. In order to make such a statement, the authors need to perform PHF14 and SIRT1 ChIP-seq experiments to show their localization at H2A.Z sites and how their localization changes when H2A.Z.1, H2A.Z.2, PHF14 or SIRT1 are depleted. Hence, I would also remove the model in Figure 7D.

We agree with the reviewer that many more experiments would be needed to formally demonstrate the original Figure 7D model, that we actually presented as a working model. Therefore, according to the reviewer suggestion, we removed the model from Figure 7D and toned down our conclusions.

Discussion: This is not the first study showing that H2A.Z.1 and H2A.Z.2 bind differentially to proteins. Draker et al. already showed a preference of BRD2 binding to H2A.Z.1, an observation also confirmed by Pünzeler et al. Further, in Pünzeler et al., PHF14, HMG20A, TCF20 and RAI1 were seen to be enriched on H2A.Z.2 nucleosomes when compared to H2A.Z.1 nucleosomes, as was also seen for other interacting proteins. The authors should rephrase their conclusions.

We agree that our phrasing was misleading (we actually forgot to include the words "using endogenous proteins" in our original manuscript). Our data are indeed the first to be performed using endogenous proteins, which is now clearly stated in the Discussion section of the revised manuscript. According to this reviewer's suggestions, we now detail Draker et al. and Pünzeler et al. results when discussing our interaction results.